# Unlimiformer: Long-Range Transformers with Unlimited Length Input

**Amanda Bertsch**  Uri Alon[*]  Graham Neubig
Matthew R. Gormley
Carnegie Mellon University, USA
{abertsch,ualon,gneubig,mgormley}@cs.cmu.edu

## Abstract

Since the proposal of transformers (Vaswani et al., 2017), these models have been limited to bounded input lengths, because of their need to attend to every token in the input. In this work, we propose Unlimiformer: a general approach that wraps any existing pretrained encoder-decoder transformer, and offloads the cross-attention computation to a single $k$-nearest-neighbor ($k$NN) index, while the returned $k$NN distances are the attention dot-product scores. This $k$NN index can be kept on either the GPU or CPU memory and queried in sub-linear time; this way, we can index practically unlimited input sequences, while every attention head in every decoder layer retrieves its top-$k$ keys, instead of attending to every key. We evaluate Unlimiformer on several long-document and book-summarization benchmarks, showing that it can process even 500k token-long inputs from the BookSum dataset, without any input truncation at test time. We demonstrate that Unlimiformer improves pretrained models such as BART (Lewis et al., 2020a) and Longformer (Beltagy et al., 2020) by extending them to unlimited inputs without additional learned weights and without modifying their code. Our code and models are publicly available, and support LLaMA-2 as well[2].

## 1 Introduction

Transformers (Vaswani et al., 2017) have risen as the dominant sequence-to-sequence architecture. Pretrained transformers generally have a context window of 512 (e.g. BERT (Devlin et al., 2019), T5 (Raffel et al., 2020)) or 1024 tokens (e.g. BART (Lewis et al., 2020b)), which are sufficient lengths for many current conditional generation datasets (XSum; Narayan et al., 2018) (CNN/DM; Nallapati et al., 2016). To address inputs between 1024 and 16,384 tokens, specialized long-context models sparsify or approximate attention (e.g. Longformer (Beltagy et al., 2020), Performers (Choromanski et al., 2020)), allowing the maximum input length to quadruple while remaining computationally feasible. Most long-document summarization and question-answering datasets, such as SCROLLS (Shaham et al., 2022), are included in this range.

Yet tasks that involve long narratives, such as book summarization (Kryściński et al., 2021), can contain inputs *exceeding 500k tokens*. Figure 1 shows the input lengths of several popular summarization and question-answering datasets, plotted against common context window lengths; the longest inputs are more than 34 times longer than Longformer's context window.

In these extremely-long-input cases, vanilla transformers cannot be simply scaled, as naïve self-attention has quadratic complexity. Long-input transformers usually *modify the base architecture*, and thus necessitate re-pre-training the model from scratch, which requires significant computational

---

[*]Now at Google DeepMind
[2]https://github.com/abertsch72/unlimiformer

37th Conference on Neural Information Processing Systems (NeurIPS 2023).

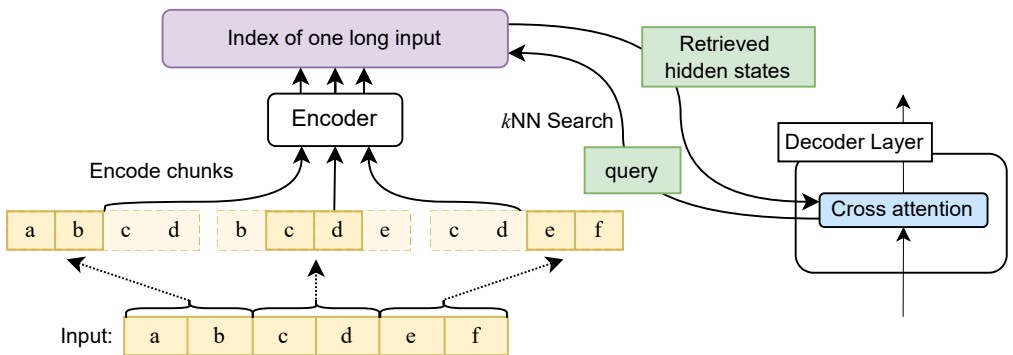

Figure 2: In this example, a given LM's encoder's maximum input length is 2 tokens. A 6-token input is encoded in chunks and indexed in an index. We inject Unlimiformer into each decoder layer prior to cross-attention. In Unlimiformer, we perform $k$NN search to select a 2-token context for each attention head from the index. This makes cross-attention attend to tokens from the entire input sequence, without adding parameters and without changing the given LM's architecture.

resources. Other architectures such as Longformer-Encoder-Decoder (LED; Beltagy et al., 2020) *can* leverage pretrained models, but they still need to further train new position embeddings or global attention weights, which is computationally and environmentally costly.

We introduce Unlimiformer, a retrieval-based approach to augment pretrained language models to accept inputs of unbounded length at test time. Given a long input sequence, Unlimiformer constructs a $k$-nearest-neighbor ($k$NN) index over the hidden states of all input tokens. Then, every standard cross-attention head in every decoder layer queries the $k$NN index, such that the $k$NN distances are the attention dot-product scores, and attends only to the top-$k$ input tokens. In preliminary experiments, we found that the top-$k$ attention keys cover more than 99% of the attention mass, and thus attending only to the top-$k$ keys is an accurate approximation of the full, exact, attention. Unlimiformer can be injected into any existing encoder-decoder transformer to permit unbounded inputs. The index can be stored in either GPU or CPU memory, needs to hold only *a single vector per input token*, and can be queried in sublinear time. Unlimiformer is illustrated in Figure 2.

Unlimiformer is a *generic* approach: it can be applied to trained models and improve existing checkpoints without adding weights and without further training. When *finetuning* Unlimiformer, performance is even further improved: across a variety of long-range datasets, not only that

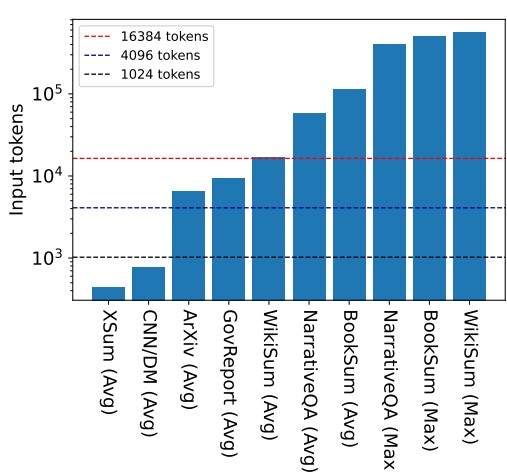

Figure 1: Long-range transformers can avoid input truncation in some datasets; however, there are datasets with inputs many times longer than these models' maximum input length. The dotted lines represent three common maximum input lengths for models; the bars are the average or maximum input length in each dataset, as indicated. Averages for datasets from Koh et al. (2022).

Unlimiformer performs better than strong long-range transformers such as LED (Beltagy et al., 2020), PRIMERA (Xiao et al., 2022), SLED (Ivgi et al., 2022) and Memorizing Transformers (Wu et al., 2022), but Unlimiformer can be applied *on top of* such models to further improve them.

## 2  Unlimiformer

Given a trained encoder-decoder transformer, Unlimiformer allows each cross-attention head to choose separate keys to attend to from the full-length input, at each decoding step. We inject a $k$NN

search into each decoder layer: prior to cross-attention, the model performs a nearest-neighbor search in a $k$NN index to choose a set of per-decoder-layer per-attention-head tokens to attend to.

## 2.1 Encoding

To encode an input sequence that is longer than the model's context window, we use the given model's encoder to encode overlapping chunks of the input, following Ivgi et al. (2022). We keep only the middle half of the encoded vectors from each chunk, to ensure that the encodings have sufficient context on both sides. Finally, we index the encoded inputs in a $k$NN index, using a library such as Faiss (Johnson et al., 2019), using dot-product as the index's nearest-neighbor similarity metric.

## 2.2 Retrieval-augmented Cross-Attention

In standard cross-attention, a transformer decoder attends to the encoder's top-layer hidden states, where the encoder usually truncates the input and encodes only the $k$ first tokens in the input sequence.

Instead of attending only to this $k$-token prefix of the input, we retrieve the top-$k$ hidden states from the $k$NN index for each cross-attention head, and attend *only to these top-$k$*. This allows retrieval from the *entire* input sequence instead of truncating. Our approach is also cheaper, in computation and GPU-memory, than attending to all input tokens; and because softmax is dominated by the largest values, retrieving the most-attended tokens preserves the vast majority of attention mass.

Figure 2 illustrates our generic changes to any sequence-to-sequence transformer's architecture. The full input is encoded using the encoder in chunks and indexed in a $k$NN index; then, the index of encoded hidden states is queried at each decoding step. The $k$NN search step is non-parametric and can be injected into any pretrained seq2seq transformer. The search step reformulates attention for space efficiency, as detailed below.

## 2.3 Attention reformulation

Let $\boldsymbol{h}_d$ be the decoder hidden state and $\boldsymbol{h}_e$ be an encoder's last layer hidden state. The standard cross-attention computation for a single head in a transformer is:

$$\text{Attn}(Q, K, V) = \text{softmax}\left(\frac{QK^T}{\sqrt{d_k}}\right) V \tag{1}$$

where $Q = \boldsymbol{h}_d W_q$ is the product of the decoder states $\boldsymbol{h}_d$ and the query weight matrix $W_q$; the keys $K = \boldsymbol{h}_e W_k$ are the product of the last encoder hidden states $\boldsymbol{h}_e$ with the key weight matrix $W_k$; and $V = \boldsymbol{h}_e W_v$ is similarly the product of $\boldsymbol{h}_e$ with the value weight matrix $W_v$. Our goal is to retrieve a set of keys $K_{best}$ that maximize $QK_{best}^T$, with the size of $K_{best}$ fixed to the size of the model's context window, and then compute the standard attention over $K_{best}$ only.

Note that the linear layers $W_q$, $W_k$, and $W_v$ are layer-specific and head-specific. Thus, naïvely creating an index from the keys $K = \boldsymbol{h}_e W_k$ and querying this index using the query vectors will require constructing separate indexes for the keys and values at each layer and each head, for a total of $2 \times L \times H$ indexes, where $L$ is the number of decoder layers and $H$ is the number of attention heads. In fact, this exact naïve approach was taken by Memorizing Transformers (Wu et al., 2022), who pioneered the use of a $k$NN index for previously encoded inputs.[3] A separate index for each attention head in each decoder layer is both time-intensive to create and space-intensive to store. So, not surprisingly, Wu et al. (2022) apply their memory layer to only a *single* decoder layer.

Instead, we present a different order of computing the well-known transformer attention formula, which allows us to store a *single* index across *all* attention heads and all decoder layers, *without changing* the mathematical definition of the transformer's standard dot-product attention. The

---

[3] See Memorizing Transformers' official implementation at https://github.com/google-research/meliad/blob/main/transformer/memory_factory.py#L78-L79 and https://github.com/google-research/meliad/blob/main/transformer/memory_layer.py#L334-L339

dot-product part of the transformer's attention computation can be rewritten as follows:[4]

$$QK^T = (\boldsymbol{h}_d W_q)(\boldsymbol{h}_e W_k)^\top \tag{2}$$
$$= (\boldsymbol{h}_d W_q) W_k^\top \boldsymbol{h}_e^\top$$
$$= (\boldsymbol{h}_d W_q W_k^\top) \boldsymbol{h}_e^\top$$

Thus, the retrieval step can be formulated as choosing the encoder hidden states $\boldsymbol{h}_e$ that maximize $(\boldsymbol{h}_d W_q W_k^\top) \boldsymbol{h}_e^\top$. This rewriting has two major advantages: first, *there is no need to index the keys for each head and layer separately*: we can create a *single* index of the hidden states $\boldsymbol{h}_e$ only, and just project the queries to $\boldsymbol{h}_d W_q W_k^\top$ using head-specific and layer-specific $W_q$ and $W_k$; second, the *values* can be calculated trivially given $\boldsymbol{h}_e$, so there is no need to store the values in a separate index from the keys before decoding. Thus, instead of constructing $2 \times L \times H$ indexes and retrieving from all indexes during each decoding step, we construct a *single* index from $\boldsymbol{h}_e$ and retrieve from it by just projecting the decoder hidden states to per-head per-layer $\boldsymbol{h}_d W_q W_k^\top$.

Using our reformulation, the index stores only a *single* vector per input token. Using 16-bit floats and hidden states of size 1024, this requires only 2GB of memory for 1,000,000 input tokens. Since indexes can be offloaded to the CPU memory, Unlimiformer's input length is practically unlimited.

## 3  Training Unlimiformer

Unlimiformer can be used, at test time, with an already-trained model, and lead to gains without further training, as we show later in Table 3. Next, we turn our focus to training approaches to further improve the performance of Unlimiformer. Table 1 summarizes and contrasts the training approaches described below, and Appendix A contains further implementation details.

| Method name | Training input | total # tokens in example seen at training time | Validation input (early stopping) | Test input |
|---|---|---|---|---|
| Baseline | 1024 | 1024 | 1024 | 1024 |
| +test Unlimiformer | 1024 | 1024 | 1024 | unlimited |
| +early stop w/ Unlimiformer | 1024 | 1024 | unlimited | unlimited |
| Train chunked +test Unlimiformer | 1024 | all | unlimited | unlimited |
| SLED (Ivgi et al., 2022) | 16k | 16k | 16k | 16k |
| Longformer (Beltagy et al., 2020) | 16k | 16k | 16k | 16k |
| Random-encoded training | 8-16k | 8-16k | unlimited | unlimited |
| Retrieval training | 8-16k | 8-16k | unlimited | unlimited |
| Alternating training | 8-16k | 8-16k | unlimited | unlimited |

Table 1: A comparison of the training approaches using BART (context window size 1024) as a running example. The dashed line separates methods that are approximately the same training-time cost as the baseline, from those that require significant additional compute.

### 3.1  Low (additional-) Cost Training Methods: Applying Unlimiformer at validation or test-time only

We first consider training approaches that do not require significant additional compute as compared to the standard finetuning regime.

**+test Unlimiformer:** As the simplest case, we use a standard fine-tuning regime, where the input is truncated during training. At inference time only, we inject Unlimiformer into the trained model to process full-length inputs.

**+early stop w/ Unlimiformer:** We train without Unlimiformer, but when we evaluate the model for early stopping, we use Unlimiformer for generation on the validation set. This results in choosing a slightly different checkpoint to stop training at; the additional computational cost here is minor, and comes only from the application of Unlimiformer over the validation set.

---

[4]For brevity, we omit the linear layers' bias term, as softmax is invariant to constants added to all inputs.

***Train chunked +test Unlimiformer:*** As a data augmentation approach, we split each training example into non-overlapping chunks of the context-window size, and treat each chunk as its own training example. Then, we finetune the model as normal, with this augmented set of examples as the training data. This is orthogonal to the Unlimiformer model, but has the advantage that all tokens from the full-length training example are observed during training instead of truncated—albeit across several examples. We apply early stopping with Unlimiformer on the validation set; when validating, we do not chunk inputs.

## 3.2 Long-range Training Methods: Applying Unlimiformer at training time

We also consider training Unlimiformer directly, which introduces additional computational cost.

***Random-encoded training:*** At each training step, the full (longer-than-context-window) training example is encoded in chunks; then, the keys for each decoder layer are chosen randomly from the encoded hidden states. This weakly simulates a nearest-neighbors search, but is computationally cheaper.

***Retrieval training:*** At each training step, the keys for each decoder head and layer are selected using a $k$NN search. When inputs are longer than 16k tokens, we truncated the input to 16k tokens at training time due to GPU memory requirements. This training approach is the closest to the test-time computation.

***Alternating training:*** In this approach we alternate batches of *Random-encoded training* and *Retrieval training*. *Retrieval training* is identical to the test-time setting, while *Random-encoded* introduces regularization that makes the model attend to non-top-$k$ keys as well.

## 4 Experimental Settings

### 4.1 Datasets

| Dataset | Domain | # examples | Avg # tokens Input | Output | Input length distribution | |
|---|---|---|---|---|---|---|
| GovReport | Government | 19,402 | 9,616 | 597 | 74 | 303192 |
| SummScreen | TV shows | 4,348 | 8,987 | 137 | 2365 | 22635 |
| BookSum | Literature | 436 | 143,301 | 1294 | 8388 | 642376 |

Table 2: Dataset statistics. The last column is a visualization of the distribution of input example lengths in each dataset; the histogram is binned by powers of 2, with the minimum and maximum input size displayed on either end. The dotted line indicates the mean length.

We experiment with two long-document- and one book-summarization datasets from varying domains. Table 2 summarizes statistics for each dataset. GovReport and SummScreen were taken from the SCROLLS benchmark (Shaham et al., 2022). **GovReport** (Huang et al., 2021) is a long-document summarization dataset where the task is to write the executive summary of a US government report. **SummScreen** (Chen et al., 2022) is a long-document summarization dataset where the task is to write the recap of a TV show episode (such as "Friends"), given the transcript of the entire episode. **BookSum** (Kryściński et al., 2021) is a book-summarization dataset of entire books. BookSum has paragraph, chapter, and book-level settings; we consider the hardest BOOKSUM-Book setting, where the task is to generate a book-level summary given the full text of the novel as input.

**Metrics**  We report ROUGE 1/2/L (Lin, 2004) and BERTScore F1 (Zhang et al., 2019). Following Zhang et al. (2021), in BookSum we also used Entity Mention Recall ("EntMent") as a proxy for the informativeness of the candidate summaries. EntMent measured the fraction of gold entities mentioned in the candidate summary. Additional evaluation details are provided in Appendix C.

### 4.2 Baselines

**BART** (base) (Lewis et al., 2020b) is a pretrained seq2seq model (139M parameters), commonly used for summarization tasks. Its maximum input sequence length is 1024 tokens.

| Base model | Training method | ROUGE 1 / 2 / L / BERTScore | |
| --- | --- | --- | --- |
| | | GovReport | SummScreen |
| BART$_{base}$ | Standard finetuning | 48.7 / 19.2 / **22.8** / 64.3 | 29.7 / 6.2 / 17.7 / 56.3 |
| BART$_{base}$ | +test SLED (Ivgi et al., 2022) | 45.8 / 16.1 / 20.2 / 62.7 | 27.5 / 5.5 / 16.7 / 55.9 |
| BART$_{base}$ | +test Unlimiformer | 49.7 / 19.6 / 22.0 / 64.8 | 30.9 / 6.5 / 18.2 / 57.5 |
| BART$_{base}$ | +early stop w/ Unlimiformer | **51.0 / 20.5** / 21.5 / **65.1** | **32.1 / 6.8 / 18.6 / 57.6** |
| BART$_{base}$ | Train chunked | 46.2 / 17.8 / 21.7 / 63.3 | 28.1 / 5.6 / 17.0 / 55.6 |
| BART$_{base}$ | +test Unlimiformer | **53.4 / 22.5 / 22.5 / 66.0** | **29.3 / 6.6 / 17.6 / 57.0** |
| PRIMERA | Standard finetuning | 55.1 / 23.9 / 25.9 / 67.0 | 32.3 / 7.1 / 18.3 / 57.1 |
| PRIMERA | +test Unlimiformer | **56.5 / 24.8 / 26.3 / 67.7** | **33.3 / 7.7 / 19.1 / 57.6** |

Table 3: Results on long-document summarization, low-cost training methods: the training costs are no higher than standard finetuning that truncates the inputs to the model's max input size. The best metric in every training category is marked in **bold**. PRIMERA (Xiao et al., 2022) is a Longformer-Encoder-Decoder (Beltagy et al., 2020) with additional summarization-specific pretraining.

**PRIMERA** (Xiao et al., 2022) is a Longformer-Encoder-Decoder (LED$_{large}$; Beltagy et al., 2020) (447M parameters), pretrained specifically for multi-document summarization, with maximum input length of 4096 tokens.

**SLED** (Ivgi et al., 2022) extends encoder-decoder models for longer contexts by applying fusion in-decoder (Izacard and Grave, 2021): the long input is encoded in chunks, and the decoder then attends to *all* input tokens. This allows the use of pretrained models, albeit with expensive fine-tuning. The input sequence length is eventually memory bounded.

**Memorizing Transformers** (Wu et al., 2022) is the most similar work to ours; they propose extending a transformer with a trainable attention gate that moderates between the standard cross-attention and attention over retrieved keys from a datastore. Since their public implementation[5] is "not officially supported" and is not fully reproducible, we approximated it by using attention over the index in only a *single* decoder layer; this is equivalent to their setting with the learned interpolation parameter $g$ set to 1.[6] Our work differs from Memorizing Transformers in several key ways: Wu et al. (2022) added additional weights, and thus cannot easily leverage pretrained LMs, while Unlimiformer is fully non-parametric and can improve performance without fine-tuning; further, Wu et al. (2022) applies retrieval attention to only a *single* layer because of computational constraints, while our attention reformulation enables the use of Unlimiformer in *every* decoder layer with individualized retrieval per-head, while still being more efficient than Memorizing Transformers, as we detail in Section 2.3.

## 5 Results

### 5.1 Long Document Summarization

**Low-cost training** Table 3 shows the results in the long-document summarization datasets. First, we can see that applying Unlimiformer on an existing checkpoint without any training (+*test Unlimiformer*) improves BART$_{base}$ by, for example, 1.8 ROUGE-1 points on both datasets, and improves PRIMERA by 1-1.4 ROUGE-1 points. In contrast, without additional training, SLED decreases performance. Thus, Unlimiformer is the only model that can provide benefits without further training.

*Early stop w/ Unlimiformer* further improves the base model without any special training: it provides, for example, 3.3 ROUGE-1 points gain on GovReport, while the training computational cost is identical to standard finetuning. *Train chunked* does not provide benefits on its own; however injecting Unlimiformer applied at test time results in the most significant gains: 7.2 ROUGE-1 and 3 BERTScore points improvements, while training is as computationally cheap as standard finetuning.

---

[5] https://github.com/google-research/meliad

[6] Wu et al. (2022) note that in their experiments that most heads learned a value for $g$ such that they attended "almost exclusively" to the external memory.

| Base model | Training method | ROUGE 1 / 2 / L / BERTScore | |
| | | GovReport | SummScreen |
| --- | --- | --- | --- |
| BART$_{base}$ | Standard finetuning | 48.7 / 19.2 / 22.8 / 64.3 | 29.7 / 6.2 / 17.7 / 56.3 |
| BART$_{base}$ | SLED (Ivgi et al., 2022) | 54.7 / 24.4 / 25.4 / 67.0 | 32.7 / 7.9 / 19.1 / 58.4 |
| BART$_{base}$ | Memorizing transformers | 55.2 / 25.1 / 26.4 / 67.5 | 32.7 / 7.4 / 19.2 / 57.4 |
| BART$_{base}$ | Unlimiformer (this work) | **56.6 / 26.3 / 27.6 / 68.2** | **34.7 / 8.5 / 19.9 / 58.5** |
| PRIMERA | Standard finetuning | 55.1 / 23.9 / 25.9 / 67.0 | 32.3 / 7.1 / 18.3 / 57.1 |
| PRIMERA | Memorizing transformers | 57.0 / 25.3 / 26.5 / 67.7 | 33.0 / 7.3 / 18.4 / 57.3 |
| PRIMERA | Unlimiformer (this work) | **57.4 / 26.2 / 28.0 / 68.1** | **33.3 / 7.6 / 18.9 / 57.7** |

Table 4: Test results on long-document datasets, when allowing compute-costly, long-range training methods, using different base models. The best metric in every dataset and every training category is marked in **bold**. The Unlimiformer results in this table are from using the *alternating training* strategy.

| Base model | Training method | ROUGE 1 / 2 / L | EntMent |
| --- | --- | --- | --- |
| BART$_{base}$ | Hierarchical (Kryściński et al., 2021) | 30.0 / 6.0 / 11.0 | - |
| BART$_{base}$ | Standard finetuning | 36.4 / 7.6 / 15.3 | 10.0 |
| BART$_{base}$ | +test Unlimiformer | 35.5 / **7.7** / 15.4 | **21.9** |
| BART$_{base}$ | +early stop w/ Unlimiformer | 35.5 / **7.7** / 15.4 | **21.9** |
| BART$_{base}$ | Memorizing Transformers | 35.6 / 6.4 / 14.6 | 10.1 |
| BART$_{base}$ | Unlimiformer (retrieval training) | 36.8 / 8.3 / 15.7 | 20.3 |
| BART$_{base}$ | Unlimiformer (random-encoded training) | **37.3** / 6.7 / 15.2 | 20.8 |
| BART$_{base}$ | Unlimiformer (alternating training) | 36.7 / 7.3 / **15.5** | 20.3 |
| PRIMERA | Standard finetuning | 38.6 / 7.2 / 15.6 | 11.6 |
| PRIMERA | +test Unlimiformer | 38.3 / 7.5 / 15.9 | 18.9 |
| PRIMERA | +early stop w/ Unlimiformer | **39.5** / 7.3 / 15.8 | 22.2 |
| PRIMERA | Unlimiformer (retrieval training) | 37.9 / **8.2 / 16.3** | **25.5** |
| PRIMERA | Unlimiformer (random-encoded training) | **39.5** / 7.1 / 15.9 | 19.7 |
| PRIMERA | Unlimiformer (alternating training) | 38.2 / 7.1 / 16.0 | 23.4 |

Table 5: Results on BookSum (average input length ≈ **143k** tokens). *EntMent* is entity recall. Hierarchical summarization is a baseline reported by Kryściński et al. (2021), where chapter summaries are condensed to form a book summary. The best metric in every dataset is marked in **bold**.

**Long-range training** Table 4 shows results when allowing computationally expensive training approaches. As shown, in almost all metrics and datasets, Unlimiformer outperforms the SLED and Memorizing Transformers baselines when using the same base model.

The PRIMERA experiments in Table 4 highlight two important points: first, Unlimiformer+BART$_{base}$ performs better than the base PRIMERA across all metrics and datasets, even though PRIMERA is larger and was pretrained on much more data, using a pretraining objective that was designed for summarization; second, not only can Unlimiformer outperform Longformer-based models such as PRIMERA, Unlimiformer can also be applied *on top* of existing long-range transformers and further improve them: Unlimiformer+PRIMERA improves over PRIMERA across all metrics and datasets. Additional results on the validation set are provided in Appendix E.

## 5.2 Book Summarization

Table 5 shows the result on BookSum. As shown, Unlimiformer improves both base models BART$_{base}$ and PRIMERA, in both low-cost training approaches such as *Early stop w/ Unlimiformer*, as well as in the long-range training approaches. *Random-encoded-*, *Retrieval-*, and *Alternating-* training show competitive performance, with the best method varying across datasets and models.

We found that although Unlimiformer outperforms all base models on BookSum (Table 5), the base BART (*Standard finetuning*, which truncates the input to the first 1024 tokens) shows competitive ROUGE and BERTScore metrics. This is strongly counterintuitive for book summarization, where

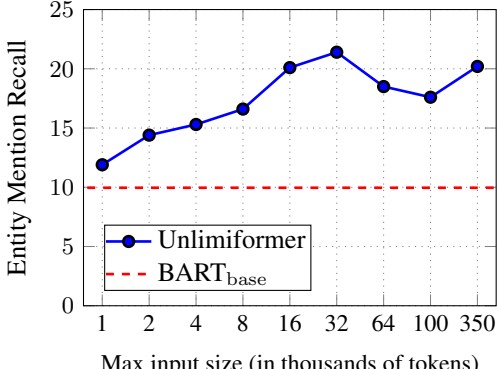
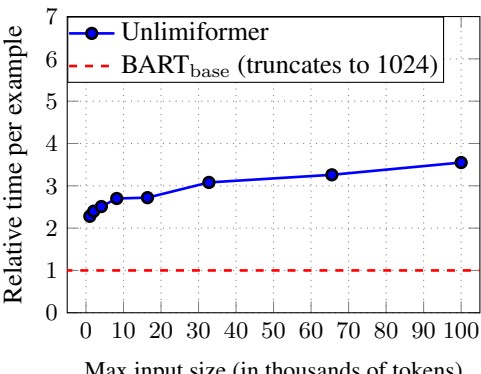

Figure 3: As the maximum datastore size increases, the entity recall generally increases. At all datastore sizes, Unlimiformer outperforms the baseline (BART, in red).

Figure 4: As the maximum datastore size increases, the inference cost increases sublinearly. The plot shows total wall-clock inference time per example.

the book's plot should not be apparent from reading only the first pages. In the outputs from this base model, we observe limited coherence and a high rate of hallucination (see Appendix F for an example with analysis). However, this is not reflected in n-gram-based overlaps, and BERTScore does not strongly distinguish between any of the BookSum models.

Nonetheless, the ability to attend to unlimited inputs at test time allows Unlimiformer to achieve significantly better Entity Mention Recall (EntMent): the Unlimiformer models exhibit far higher EntMent, and even adding Unlimiformer only at test time without costly training (*Early stop w/ Unlimiformer*) *doubles* the entity recall compared to the base model. Further, Unlimiformer improves EntMent in the base PRIMERA from 11.6 to 25.5 in Unlimiformer+PRIMERA.

## 6 Analysis

**Is the long input really needed?** As found in various recent papers (Shaham et al., 2022; Kedzie et al., 2018), many text generation datasets do not require long-range modeling, since most of the needed information is concentrated at the beginning of the input. To evaluate whether Unlimiformer really utilizes long inputs, we experimented with limiting the input length in BookSum. Figure 3 shows the performance of Unlimiformer in BookSum: EntMent increases almost monotonically with input length, suggesting Unlimiformer exploits the longer inputs to generate better outputs.

Other work (Jiang and Bansal, 2019) has found that in some datasets, the needed information is concentrated in only part of the input, which is not necessarily the beginning. We observed this trend in WikiSum, a multi-document summarization dataset where the inputs are all references of a Wikipedia article and the output summary is the intro paragraph of the article (Liu* et al., 2018)[7]. As a strong baseline, we followed Liu* et al. (2018), and ranked the input paragraphs according to TF-IDF. Unlimiformer did not improve over a baseline that uses only the first 1024 tokens of this sorted input, suggesting that the full input is not necessary to produce the summary on this dataset[8].

**Computational cost** Although Unlimiformer does not introduce additional trained parameters, the encoding of the full input, index construction, and index search increase the processing time during both training and inference. We plot the computational cost of inference with respect to the input length in Figure 4. When all inputs are restricted to 1,024 tokens, Unlimiformer requires a small additional time overhead relative to the baseline for indexing and search. However, the benefits of

---

[7]A full copy of WikiSum is not available online; details of our scraped copy are in Appendix B.

[8]It is also possible that the baseline's strong performance on this task is due to some contamination from pretraining data; BART was trained on books and Wikipedia data, which likely includes some articles used in WikiSum. Any contamination from pretraining here strengthens the baseline performance.

| Base model | Training method | QASPER F1 | Contract NLI Exact Match | QMSum ROUGE 1 / 2 / L | Narrative QA F1 |
|---|---|---|---|---|---|
| BART$_{base}$ | Standard finetuning | 22.0 | 77.5 | 30.8 / **8.7** / **20.8** | 15.5 |
| BART$_{base}$ | Unlimiformer | **27.5** | **77.7** | **30.9** / 8.0 / 19.9 | **18.5** |

Table 6: Results on question answering, query-based summarization, and NLI datasets.

Unlimiformer are clear as input length increases: the total GPU-time required increases *sublinearly* with input length[9]. Additional GPU-time measurements are reported in in Appendix D.

**Performance on other tasks** We measure the performance of Unlimiformer relative to the base model on 4 additional datasets: QASPER (Dasigi et al., 2021), a question-answering dataset over NLP papers; Contract NLI (Koreeda and Manning, 2021), a natural language inference dataset over legal contracts; QMSum (Zhong et al., 2021), a query-based summarization dataset over meeting transcripts; and NarrativeQA (Kočiský et al., 2018), a reading comprehension dataset over narratives[10].

Table 6 shows the performance of BART-Unlimiformer (with alternating training) relative to base BART. On three of the four datasets, applying Unlimiformer improves over the base model.

**What is attended to?**

We plotted the frequency of retrieval for keys across the full decoding process for the test set of BookSum, the dataset with the longest inputs. The average number of input embeddings retrieved at least once varied by method, from 43.5% of all tokens for the test-time-only Unlimiformer to 64.5% of all tokens for the alternating-training model[11].

Figure 5 shows the retrieval locations for the alternating-training model. We found no specific skew or pattern in the retrieved keys, and keys from the entire input were used by the model; for all models, the median location of a retrieved key was between 49.73% and 49.87% of the way through the input document.

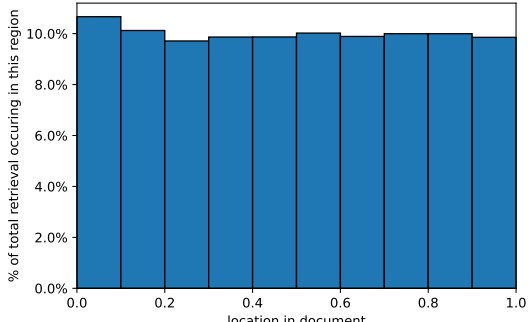

Figure 5: Histogram of location of retrieved embeddings in the original document (averaged over the BookSum test set). There is a slight bump at the beginning (first 10% of the book), but otherwise no strong trend, with tokens retrieved quite uniformly from the entire inputs.

## 7  Related Work

**Long-range transformers** Previous long-range transformers change the transformer architecture to reduce its space or time requirements (Tay et al., 2020). Most solutions achieve this reduction through sparsifying the attention mechanism (Child et al., 2019; Kitaev et al., 2020; Beltagy et al., 2020; Roy et al., 2020; Ainslie et al., 2020; Zaheer et al., 2020). Other works approximate or replace the attention mechanism entirely (Wang et al., 2020; Katharopoulos et al., 2020; Choromanski et al., 2020; Lee-Thorp et al., 2021). All these approaches change the standard transformer architecture or its training objective (Zhong et al., 2022), and thus require pretraining the model from scratch, which does not allow to leverage existing pretrained models. In contrast, Unlimiformer is *generic*, can be injected into any encoder-decoder transformer, and improve it either without training or with merely fine-tuning. This way, Unlimiformer can leverage any already-pretrained model.

**Comparison to Wu et al. (2022)** The closest work to ours is Memorizing Transformers (Wu et al., 2022). Memorizing Transformers construct two datastores for each attention head in each layer,

---

[9]The time to encode the input increases linearly and the time to decode increases sublinearly; because the encoding is only a small part of the total time during inference, this results in an overall sublinear trend.

[10]We report results on the validation split from SCROLLS Shaham et al. (2022).

[11]Note that this is not the percentage of the input that *influenced* the output; we retrieve encoded, contextualized hidden states, so even vectors that were not directly retrieved by the decoder impacted the final output.

and due to memory constraints can thus apply their approach only to a single decoder layer. In contrast, thanks to our attention reformulation (Section 2.3) Unlimiformer can use a *single* index for all decoder layers, and thus allow *all* cross-attention heads in all decoder layers retrieve from the long context. As we show in Section 5, this results in significant empirical gains over retrieving only at a single layer. Further, Memorizing Transformers introduce additional learned weights, thus they *must* be trained to incorporate their memory, and thus cannot easily leverage pretrained models; as we show in Section 5 Unlimiformer can improve existing models without any training, and thus can be applied to any existing transformer. Additionally, Memorizing Transformers focused on decoder-only models; while our approach could also be applied to decoder-only models (and would provide a space efficiency boost there as well), we focus on encoder-decoder models in this work.

**Comparison to Ivgi et al. (2022)** Another related work to ours is SLED (Ivgi et al., 2022). SLED encodes long inputs in chunks, similarly to Unlimiformer, but the decoder in SLED attends to *all* inputs at the same time. This in practice limits SLED to only about 16k token-long inputs on a single GPU; in contrast, instead of attending to *all* input tokens, Unlimiformer attends only to the top-$k$ input tokens for every attention head, and thus can process unlimited inputs in practice, while preserving more than 99% of the attention mass. Further, SLED requires computationally costly training, while Unlimiformer can provide benefits without any training.

Additional related work is discussed in Appendix G.

# 8   Conclusions

We present Unlimiformer, an approach for augmenting pretrained encoder-decoders and offloading the cross-attention computation to a $k$NN index, to allow for unlimited length input. Instead of attending to all keys, this $k$NN index allows every cross-attention head in every decoder layer to retrieve and attend only to its top-$k$ keys. We evaluate Unlimiformer on several long-document and book-summarization benchmarks having inputs of up to 500K tokens, and show that Unlimiformer improves existing models, even without further training. When *training* with Unlimiformer, not only that Unlimiformer makes smaller models such as BART perform better than larger Longformer-based models, Unlimiformer can be applied on top of Longformer-based models and further improve them.

Many real-world NLP tasks require processing large amounts of data or text. Yet pretraining large models incurs substantial carbon costs (Strubell et al., 2019), which increase with the length of the context window; by choosing instead to modify already-pretrained models to process longer inputs, we aim to gain the benefits of long contexts with less computational cost. We hope that our approach will allow the democratization of long-range transformers, especially for researchers and practitioners with low-compute resources. Toward this end, we release our code at https://github.com/abertsch72/unlimiformer. Our code is based on HuggingFace Transformers (Wolf et al., 2020), without changing any individual architecture's code, and thus can be injected into any encoder-decoder model, and supports decoder models such as LLaMA-2 as well.

# 9   Limitations

In our experiments, we have only considered English-language datasets. While we have no reason to believe the method would suffer from the use of a different high-resourced language, the quality of the nearest-neighbors search depends on the quality of the indexed keys.

The length of inputs that can be used at training time is limited by the GPU memory, as the embeddings and their computational graph must be stored for backpropagation. Multi-GPU training would allow longer inputs at training time.

At inference time, Unlimiformer can process the longest inputs when the index is offloaded to the CPU memory. In this case, Unlimiformer requires to index only a single vector per input token, which practically means unlimited inputs for any modern server and even small machines during inference. However, offloading the index to the CPU results in higher test-time latency compared to storing the encoded hidden states and the index on the GPU. In our experiments, we were able to use a GPU index for input examples exceeding 500k tokens (on GPUs no larger than 48 GBs), but this may be a concern when using smaller GPUs or larger models.

## Acknowledgments

We are grateful to Sireesh Gururaja for useful feedback on a draft of this paper. We also thank Maor Ivgi for the help in reproducing results from SLED (Ivgi et al., 2022) and for sharing code and models, and Uri Shaham for the discussions about the SCROLLS benchmark (Shaham et al., 2022). We are also grateful to the anonymous reviewers for their useful comments and suggestions.

This work was supported in part by grants from 3M — M*Modal and from the National Science Foundation Graduate Research Fellowship Program under Grant No. DGE2140739. Any opinions, findings, and conclusions or recommendations expressed in this material are those of the authors and do not necessarily reflect the views of the sponsors.

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

## A  Training details

At training time, we must backpropagate through the operations described above. Thus, the input length is bounded more strictly – the number of tokens in the full input must fit in GPU memory while the model is loaded. For the computationally expensive methods, we train using batch size 1 and truncate the longest inputs (generally, to 16k tokens). At test time, we use the full input without truncation. We train one model per setting, using the hyperparameter settings from SLED (Ivgi et al., 2022) and early stopping.

## B  WikiSum scraping

We rescraped the dataset, following the same preprocessing steps as the original authors. We observe that many inputs in the scraped dataset are shorter than reported, likely due to changes in availability of the data since 2017; as a preprocessing step, we remove all inputs that are less than 1457 words, which is the 40th percentile of citation size for the original dataset. We trained on 10,000 randomly selected examples from this version of WikiSum and evaluate on 2,000 randomly sampled examples (1,000 validation, 1,000 test), maintaining the same sample across all experiments. When sampling, we respect the original WikiSum train/validation/test split. We release the subset we trained on as well as our modified version of the scraping code.

## C  Evaluation details

Vanilla BERTScore is only well-defined up to 512 tokens; for GovReport and ScriptSumm, we evaluate using `facebook/bart-large-mnli` instead. This model has context size 1024. For BookSum, we experimented with using `allenai/longformer-large-4096` (context size 4096), as many references are longer than 1024 tokens; however, we found that this approach had no distinguishing power between model outputs, ranking all models tested within 0.3 points of each other despite observing significant differences with ROUGE, EntMent, and manual inspection.

For computing Entity Mention Recall (EntMent), we used SpaCy[12] to tag all named entities in the gold summary and collected a set of unique entities. We then tagged each candidate summary and computed the percentage of entities present in this summary, that is, recall of unique entities. For the named entity recognition in EntMent, we used SpaCy's `en_core_web_lg` model.

## D  Computational Cost

We estimate the total GPU time for results presented in this paper did not exceed approximately 116 days of time on a single 48-GB A6000. The longest-training models, SLED and retrieval training for GovReport, took approximately 10 days to train.

**GPU-time**   Table 7 shows the relative cost for each method. The Unlimiformer training methodologies are higher cost than the base training; however, the largest difference occurs during inference, where the full input (in Booksum, an average of 112,885 tokens) must be encoded, instead of the 1,024 tokens encoded in the baseline approach.

Using a CPU datastore is many times slower than a GPU datastore because of slower search and the need to transfer retrieved embeddings to the GPU. In our experiments, we were able to use a GPU datastore for input examples exceeding 500k tokens (on GPUs no larger than 48 GBs), but this may be a concern when using smaller GPUs or even larger inputs. Additionally, CPU indices are necessary for models with context windows larger than 2048 tokens, as the Faiss GPU index implementation does not support retrieving more than 2048 nearest neighbors; however, the datastore can still be stored on GPU.

---

[12]https://spacy.io

| Method | Relative GPU-time |
|---|---|
| Baseline training | $1.00 \pm 0.00$ |
| Chunked training | $1.02 \pm 0.02$ |
| +early stop w/ Unlimiformer | $1.00 \pm 0.00$ |
| Retrieval training | $1.89 \pm 0.06$ |
| Random-encoded training | $2.87 \pm 0.28$ |
| Baseline inference | $1.00 \pm 0.00$ |
| Unlimiformer inference | $4.48 \pm 0.56$ |

Table 7: Computational effort per epoch for different training methodologies, relative to the baseline of standard finetuning and inference. All are averaged over 3 runs on BookSum using a single 48 GB A6000 GPU, 32 GB RAM, and 16 CPUs.

| Number of layers using Unlimiformer | Memory consumption (GB) |
|---|---|
| 0 (normal inference) | 1.61 |
| 1 | 7.33 |
| 2 | 7.32 |
| 3 | 7.36 |
| 4 | 7.32 |
| 5 | 7.33 |
| 6 (all) | 7.35 |

Table 8: Memory consumption for applying Unlimiformer at different numbers of layers in BART.

**Memory usage**    Table 8 shows the memory required to apply Unlimiformer on varying numbers of layers in a BART model. Using Unlimiformer requires more memory than the base model for two reasons: index construction and the additional input processed. There is a slight overhead for constructing and storing the index. But more crucially, the base BART ("normal inference") is truncating the input to the first 1024 tokens. When Unlimiformer is used, we process the full inputs, some of which are >500,000 tokens, and so much of the additional memory cost comes from storing the additional hidden states. However, the GPU memory consumption remains constant even when we use Unlimiformer in more layers, highlighting the scalability of Unlimiformer compared to other approaches such as Memorizing Transformers, which need to allocate more memory with every layer and every attention head.

## E   Validation Results

Table 9 shows the *validation* metrics for GovReport and SummScreen.

## F   Sample Outputs

These outputs from BookSum are summaries of *The Brothers Karamazov*, an elaborate novel about a Russian family. Neither summary is fully factually correct, but the summary from the input-truncated model hallucinates several plotlines (e.g. a lover from the Congo, the many deaths of Pavel) which are not present in the original. The hallucinations in the Unlimiformer output are more constrained; for instance, it incorrectly describes Dmitri as a "nobleman" instead of a landowner and says he has been sentenced to death instead of jail. This summary features more of the novel's characters and identifies plot details from the later parts of the book, such as Dmitri's trial.

**Gold (reference) summary:**

> The Brothers Karamazov is a family tragedy centered around a father and his sons. Fyodor, the eldest Karamazov, has three sons: Dmitri, Ivan, and Alyosha. Ivan and Alyosha have the same mother, but Dmitri, the oldest, has a different mother. Fyodor is a greedy landowner, a bawdy lecher, and a neglectful father. Hence, the Karamazov brothers end up growing into young men under the care of various

| Base model | Training method | ROUGE 1 / 2 / L / BERTScore | |
| --- | --- | --- | --- |
| | | GovReport | SummScreen |
| | Low-cost training methods: | | |
| BART$_{base}$ | Standard finetuning | 47.7 / 18.5 / 22.3 / 64.0 | 30.0 / 6.5 / 17.7 / 56.7 |
| BART$_{base}$ | +test SLED | 46.0 / 16.3 / 20.3 / 62.8 | 28.4 / 5.9 / 17.0 / 56.0 |
| BART$_{base}$ | +test Unlimiformer | 49.5 / 19.6 / 21.9 / 64.8 | 31.8 / 7.1 / 18.6 / 57.8 |
| BART$_{base}$ | +early stop w/ Unlimiformer | 51.0 / 20.6 / 21.6 / 65.9 | **32.5 / 7.2 / 19.9 / 57.9** |
| BART$_{base}$ | Train chunked | 48.3 / 18.1 / 22.3 / 63.8 | 29.4 / 6.3 / 17.6 / 56.8 |
| BART$_{base}$ | +test Unlimiformer | **52.9 / 22.2 / 22.4** / 65.8 | 29.4 / 6.3 / 17.6 / 56.8 |
| | Long-range training methods: | | |
| BART$_{base}$ | SLED (Ivgi et al., 2022) | 55.5 / 24.8 / 25.8 / 66.9 | 34.2 / 8.2 / 19.2 / **58.8** |
| BART$_{base}$ | Memorizing Transformers | 55.8 / 25.6 / 26.9 / 67.7 | 32.8 / 7.6 / 19.3 / 57.7 |
| BART$_{base}$ | Unlimiformer | **57.4 / 26.4 / 27.9 / 68.2** | **35.0 / 8.3 / 19.6** / 58.4 |
| | Low-cost training methods: | | |
| PRIMERA | Standard finetuning | 55.0 / 23.6 / 25.9 / 66.9 | 33.0 / 7.8 / 18.8 / 57.4 |
| PRIMERA | +test Unlimiformer | **56.4** / 24.7 / **26.4 / 67.6** | 33.1 / 7.9 / 18.7 / 57.4 |
| PRIMERA | +early stop w/ Unlimiformer | **56.4 / 25.0 / 26.4 / 67.6** | **33.5 / 8.2 / 19.3 / 57.7** |
| | Long-range training methods: | | |
| PRIMERA | Memorizing transformers | 57.0 / 25.6 / 26.8 / 67.8 | 32.9 / 7.7 / 18.5 / 57.5 |
| PRIMERA | Unlimiformer | **58.0 / 26.5 / 28.6 / 68.3** | **34.1 / 7.9 / 19.0 / 57.8** |

Table 9: Validation results on long-document datasets (average input length between 4k to 16k tokens). The best metric in every dataset and every training category is marked in **bold**.

other people. But they all have returned home to visit their father, and it is the first time they all have been together for quite some time. Dmitri has a dispute with Fyodor over his inheritance, and Alyosha, who is living in a monastery, suggests that they see Father Zossima, Alyosha's mentor. Alyosha believes that the wise old man can settle the dispute peacefully. Father Zossima is patient and kind, but Fyodor and Dmitri end up quarreling anyway. After Fyodor drives the men to frustration, they leave the monastery separately, and Alyosha worries about their family's future. Alyosha talks to Dmitri, who confesses his complicated situation with women and money. Dmitri promised to marry a girl named Katerina, and she lent him 3,000 rubles. Instead of paying it back, he spent it on another girl named Grushenka. He wants to run away with Grushenka, but he feels that he needs to pay Katerina back before he can do so. This is why he is so interested in getting the money from Fyodor. Back at Fyodor's house, Smerdyakov is talking to the Karamazovs. Smerdyakov is an epileptic servant who was adopted by Grigory and Marfa, Fyodor's other servants. He was born to a woman named Lizaveta who died in childbirth. She was the town idiot, and she lived off charity from the other townspeople. Everyone called her "Stinking Lizaveta," and when the town found out she was pregnant, they were furious at whoever could do such a thing to a helpless girl. They decided Fyodor must have been the culprit. Grigory and Marfa gave birth to a deformed child, and when they buried the child, they found Lizaveta, who had just given birth to Smerdyakov. They adopted the child immediately, and Fyodor named him. Father Zossima is dying, and Alyosha is distraught. Instead of asking Alyosha to stay with him during his last days, however, Father Zossima tells Alyosha he should leave the monastery to be with his family. His life gets even more complicated when a young crippled girl named Lise expresses that she has feelings for him. Alyosha visits Katerina, the girl who is engaged to marry Dmitri. Ivan is in love with her, but he feels that Dmitri is a better match for her. Frustrated and disgusted with his family's situation, Ivan says he is going to leave town. Alyosha sees a boy being picked on by his schoolmates, and he tries to talk to the boy, but he bites Alyosha's hand and runs away. Later, when Alyosha is bringing money to a man named Captain Snegiryov, who has been beaten by Dmitri, he recognizes the man's son. It is Ilusha, the boy who bit his hand. The family is poor, but Captain Snegiryov refuses to take the money because he feels that he needs

to earn his son's respect after being humiliated by Dmitri–and accepting charity, especially from a Karamazov, is out of the question. When Alyosha goes back to see Katerina, he finds Lise, Madame Hohlakov's daughter. The two realize that they love each other, and they decide to get married. Alyosha goes to visit Ivan, and he finds him in a restaurant. Ivan has gone there to get away from his father, and Alyosha sits down with him to have an intimate talk. Ivan tells his brother about his thoughts regarding God and the world. He recites to Alyosha a poem he has written called "The Great Inquisitor." The poem describes Christ returning to earth in the sixteenth century. The Church throws him in jail, and The Great Inquisitor explains to him that his presence is problematic for the world. The Church has spent years trying to replace the sense of freedom Christ gave man with security. He talks about how cruel the world is, especially to innocent children. After their meal, Alyosha and Ivan part ways, feeling closer than ever. Ivan sees Smerdyakov when he goes back to his father's house, and Smerdyakov tells him he is worried about Fyodor. He is worried Dmitri will come to kill him and the old man will be helpless to save himself. Ivan goes to sleep very troubled. Father Zossima is on his deathbed, and Alyosha goes to visit him. The Elder tells those around him how much Alyosha reminds him of his older brother, a boy who died when he was a youth. He talks about being a profligate youth in the army. One day, he challenged another man to a duel because of a girl. Before the duel, however, he had a change of heart. He did not shoot and, after the duel, he retired from the army and joined a monastery. He talks about how much the Bible has affected him and says that everyone should embrace the world and the people in it. He dies. Many predicted that a miracle would happen upon Father Zossima's death, but his body begins to putrefy, filling the monastery with an awful smell. This fills the other monks with doubt that Father Zossima was the saintly man they thought he was. Alyosha is shaken by the news. He goes to see Grushenka, who has sent for him, and she admits to wanting to "ruin" him. When he tells her that Father Zossima has died, however, she becomes contrite about her callousness. She says she thinks she is a wicked person, and the two comfort each other. When Alyosha leaves, he has a renewed faith in Father Zossima and his teachings because Alyosha feels how wonderful it is to love and be loved in return. Meanwhile, Dmitri has become desperate. He wants to be with Grushenka, but he wants to pay Katerina back first. He goes on an odyssey, hoping that he can depend on the charity of others. He visits a man named Samsanov, a man who used to pursue Grushenka, and he hates Dmitri. He sends Karamazov to see a surly drunk, tricking Dmitri into thinking this man may be helpful. The man is practically incoherent, however, and Dmitri goes to find Madame Hohlakov. She tells Dmitri that the only way he will find 3,000 rubles is in the gold mines. In confusion, Dmitri concludes that Grushenka has gone to visit his father, and he goes to his father's house in a rage, carrying a brass pestle. When he arrives, he does not find Grushenka, but as he is leaving, Grigory, his father's servant, thinks he has come to murder Fyodor. The two scuffle, and Dmitri hits Grigory on the head with the pestle. After determining that the man is not dead, Dmitri flees the scene and looks for Grushenka. She is with Kalganov, a former lover who had treated her poorly. Dmitri decides that he will not end up with Grushenka and decides to kill himself after seeing her one more time. He crashes her party and sits down with her gentleman friend and some other men. The situation becomes tense, and after the gentlemen make some disparaging remarks about Russians and Dmitri, Grushenka decides she does not want to be with such an insulting and vicious man. She decides that she loves Dmitri, and as the two are coming to terms with their love, the police come to arrest him for the murder of Fyodor. As the police question Dmitri, it becomes clear that the facts all support the conclusion that he did indeed murder his father, even though he did not commit the crime. He was at the scene of the crime, wielding a weapon, the night of the murder. He had said he would kill his father on several occasions. He publicly announced he was looking for 3,000 rubles and was desperate to find them, and Fyodor reportedly had an envelope with 3,000 rubles that was stolen the night of the murder. Dmitri is carried away, and very few people believe that he is innocent of Fyodor's murder. Meanwhile, Alyosha is visiting Ilusha, the boy who bit his

hand, in the hospital. The boy has fallen quite ill, and Alyosha has gotten to know many of the boy's friends, who are also visiting him. One boy, Kolya Krassotkin, is a leader among the boys. He and Ilusha were friends, but they had a falling out because Ilusha fed a pin to a dog, and Kolya did not approve of his cruelty. When Alyosha comes to visit, he and Kolya talk for quite some time. The boy looks up to this wise man about which he has heard so much from the other boys, and he wants to impress him. The two become friends, and Alyosha treats all the boys as equals. When Kolya goes in to see Ilusha, he gives him a dog as a present. He reveals that the dog is none other but the dog Ilusha gave the piece of bread with a pin in it. Kolya has nursed the dog back to health and has fully trained him as a gesture of friendship to Ilusha. The mood is dampened, however, when the doctors go in to see Ilusha. Without even saying it, everyone understands that the boy does not have much time left. Ilusha is brave, and he tries to lift the spirits of those around him. Later, Alyosha visits his brother in jail. Dmitri tells Alyosha that Ivan has concocted a plan for his escape from jail. Alyosha goes to talk to Ivan, who feels strangely guilty about his father's death. Alyosha tells his brother that he should not feel responsible for a crime that he did not commit, but Ivan stalks off angrily. He meets Smerdyakov, who tells Ivan he thinks the Karamazov brother is guilty as an accomplice to the murder. He says that Ivan wanted his father dead and left the night of the murder to try to free himself of the responsibility of protecting his father. Ivan is angry and troubled by this, and when he talks to Smerdyakov later, Smerdyakov flatly admits to hilling Fyodor. He says that Ivan's theories and ideas were the basis for his crime and that Ivan's talks with Smerdyakov basically rationalized the deed. When Ivan returns home after this meeting, he sees a devil in his room. The devil chastises him for being a wicked person with weaknesses and foibles that have led to disastrous circumstances. Alyosha bangs on the door and finds his brother in a feverish state, muttering about a devil and Smerdyakov. Alyosha stays the night with his brother to take care of him. Dmitri's trial begins. Many people from all around come to see the spectacle of the parricide trial. Dmitri has an excellent lawyer, but it is a hard case to win. The prosecution brings many witnesses who testify to seemingly damning evidence against Dmitri. The defense, however, discredits one after another of these witnesses, showing ulterior motives or mitigating circumstances. Alyosha defends his brother from the stand, and Katerina gives a moving account of Dmitri's honorable nature. Then Ivan comes into the courtroom, waving money and implicating Smerdyakov. Since he is yelling nonsense, disrupting the trial, and generally acting crazy, the court does not believe him. Suddenly, at the end of the trial, Katerina stands up again, showing a letter from Dmitri that clearly states Dmitri's intention to kill Fyodor as a last resort. She has a change of heart and no longer wants to lie to protect a man who has hurt her so much. Word comes to the courtoom that Smerdyakov has hanged himself. After final statements are made, the verdict comes back: guilty. Dmitri is sentenced to jail. Dmitri welcomes this chance to become a new man, but he does not want to be in exile in Siberia for the rest of his life; he wants to return to his home country before he dies. Ivan is still sick, and Katerina takes care of him. Alyosha visits the boys with whom he has become friends. They are sad because Ilusha has died. Alyosha passes along Father Zossima's teachings of love and understanding, and they all embrace his words, cheering him.

**BART (with input truncated to 1024) summary:**

Fyodor Pavlovitch Karamazov, a landowner in his own day, lives with his three sons, Adelaida Ivanovna and Alexey. He is the son of the landowner, Fyodor, and his wife, Adele, are the daughters of a wealthy nobleman in the district. Adele is the daughter of a nobleman, and her father, Ivanovitch, is a prominent figure in the town. She is a beautiful, intelligent, and intelligent young woman, who is destined to be a great writer. She has a great deal of ambition, and she is determined to achieve her goals. However, her ambition is short-lived. She falls in love with a handsome young man named Ivanovn, who has recently returned from a trip to the Congo. He has a lot of money, but he is also a poor man, and he has no means of earning it. In fact, he is a miserly beggar, and so he spends most of his time

trying to earn money from the poor. He also has a wife, a beautiful young girl, who he marries, and a young son, Ivan, who lives with him. He and Adele have two children, Ivan and Aleksey, and they are very happy, but they have a problem: Ivan is a poor beggar who is unable to pay his rent, and Aleksandra is a beggar. Ivan is very ill, so he is forced to sell his estate to a rich man named Vladimir. Vladimir is a rich, well-to-do man who is also rich, but who is not well-liked by the people who live in the neighborhood. Vladimir's wife, Evgeny, is also ill, and Vladimir's son, Pavel, is very sick. Vladimir has a bad case of typhus fever, which causes him to lose his job and his money. Vladimir, Pavel's younger brother, is sick, and Pavel is sent to the hospital to recover his father's condition. Pavel is also sick and dies, and Ivan is left to care for him. The next day, Pavel and Adelayna are reunited with their father, who died in a typhus epidemic in the country. Pavel's father, Vladimir, has died, and the family is now in mourning. Pavel has been unable to find a new wife, so the family decides to send him to Moscow, where he will be reunited with his father. The family is devastated, but Pavel is able to find an old friend, a young man who has just returned from the Congo, who will be able to help him. Pavel and Ivan are devastated by the loss of his father, but Ivan and Vladimir are able to make amends with Vladimir. Pavel, however, is devastated by this news, and, after a long period of mourning, he decides to return to Moscow. He returns to Moscow to find his father dead, and to find that his father has died. He then goes to Moscow and is buried in the city cemetery, where his father is buried. Pavel dies of a heart attack, and then he is buried with his mother. He was a young, beautiful, and beautiful woman who was raised by a kindly, kind, and generous family. He had a great time with his family, but his life was cut short when his father died. Pavel was left to live with his aunt and uncle in Moscow, and when he died, he was left alone with his uncle and his two sons. He died in Moscow and his mother, who had been living with him in a convent, died in the same convent. Pavel had been a poor, poor, and ill-treated man, but now he was able to raise his son and raise his own son. Pavel survived, and now he lives in Moscow with his Aunt and Uncle. He lives with their aunt and Uncle, who are very poor and poor. They live in a small house in the countryside, and their children are raised by their uncle and aunt. They have a great love for each other, but their love for one another is not reciprocated. Pavel also dies in the convent, and after a brief illness, Pavel is left with a broken heart. He dies in Moscow. His father, a wealthy man, is the father of his son, who was also a clergyman. Pavel died of a stroke in the hospital, and was buried in a nearby cemetery. He left Pavel and his family behind. Pavel tried to help Pavel, but was unable to save him, so Pavel was forced to leave Moscow. Pavel returned to Moscow in the middle of the night. He tried to save Pavel's life, but it was too late for Pavel. Pavel managed to escape, and in the end, Pavel was killed by a bullet. Pavel took Pavel's body and died of his own wounds. Pavel left Pavel to die in the forest. Pavel stayed with the family for a few years, until he died of an illness. Pavel became ill and Pavel was sent to

**Unlimiformer summary:**

The first part of the novel focuses on the question of whether or not the Russian nobleman, Dmitri Fyodorovitch, has killed his father. In the town of Ivanovna, the lieutenant-colonel of the Mushenkhanovitch is accused of the murder of his brother Ivanovitch. The lieutenant-in-command, Vasilyevitch, takes the form of a dog, and the two men–the two men and the woman who are questioned by the court-martial–murphy. The two men cry out to the God of Russia for help in their quest to save the town. The man, afraid of the wrath of the God, hands the dog a bunch of letters that are supposed to be proof of his love for his brother. The old man–the one who had killed his mother, and then found the letter–arrives. He reads it–asked the old man to forgive him for the murder and then takes the dog away. The other men, all of whom are prisoners, demand that the man confess his crime to the court. The first and most important thing they tell the court is that they love the man. The court acquits the man and sentences the man to death. The second man–an old officer of the town, Alekandrovitch–askes to tell them the same thing. The third man–in the process of confessing his crime–is Vashenka, a drunk man

who has been sent to the town to kill his father, for reasons which are not entirely clear to the people. The woman's servant, Evgenyevna, is also the one who has told the court the story of the Medvedevitch's murder, for the good old man's and the young man's love. The three men, who are separated for the first time, are laughing at the man's attempt to seduce Mitya. The young man, in the meantime, is conscripted into the town-side. He tells the court that he loves her, but he has yet to tell her the true story. The men, in this room, demand a man to kill her, and she will not betray them. The women, in their own country, are rebelling against the man who had sent them three thousand roubles, and they will not allow the man of the people to see them. They will not let the man in the town be allowed to see the man–or Dmitriovitch; he will have her husband killed him. He will not tell the people who love him. The next man, named Vashenovitch, arrives, and takes the man away. They all begin to laugh at the fact that he has succeeded in seducing and entrusting his brother Dmitri. He is then taken away to the old woman's house, where the governor-side-of-the-world, and his sister, Arkadin, is being punished. The priestesses and the baron are shocked, for they have been so virtuous and well-suited. The only thing they will be able to do is kill the priest. They threaten to burn the priestess to death, for she has been so wicked and libidinous that she has not yet seen the priest, for her husband. The priests–ostensibly convinced that she is a woman who loves the priest and has been punished for her love and for allowing the priest to marry her. The last man, Yakivitch, arrives at the house, and, after a long day of drinking and then some of the men–is killed. He and the priest are ordered to leave the town so that the priest can finally be reunited with the people of the old lady. The final man, the commander of the St. Petersburg town of Arkadina, is sentenced to death for the crime of having killed and then the lieutenant of the governor, for taking the money. The commander, the former lieutenant-delegation of the People's Army, is summarily executed, and all the men, except for the commander, have been summarily punished for their crime. The entire town is shocked and, in a very dramatic way, the priestesses plead for the forgiveness of the man, for allowing them to kill and imprison Ivan. They plead for their brother to be restored as well, for all the people they have loved, and for the priestor to tell the story.

## G  Additional Related Work

**Long-document summarization** Prior work has proposed several strategies for long-document summarization. In particular, many methods select a subsection of input to summarize using TF-IDF (Liu* et al., 2018), smaller retriever models (Liu and Lapata, 2019), or sentence similarity metrics (Bajaj et al., 2021). An orthogonal approach is to summarize chunks of the input, then combine and condense these sub-summaries into a global summary, either using vanilla transformer models (Kryściński et al. (2021), Zhang et al. (2022), (Zhang et al., 2021)) or a specialized architecture (Liu and Lapata (2019), Grail et al. (2021)). Other work has focused on expanding the amount of text that can be processed, by applying long-context transformers or developing new long-context methods (Huang et al., 2021). However, these methods all suffer from cascading errors: if the initial trimming or chunk summarization steps remove important information, there is no way to recover that information in the downstream summary.

**Retrieval-augmented transformers** Interpolating language model probabilities with nearest neighbors retrieval from an external datastore was originally proposed by Khandelwal et al. (2019). Additional work in this space has improved the selection of neighbors (Drozdov et al., 2022) or added structure to the datastore (Alon et al., 2022). Despite the shared use of retrieval, all these works retrieve from an *external* datastore, while Unlimiformer retrieves from a single input example, independently from external cumbersome sources. Borgeaud et al. (2022) incorporate retrieval from the external datastore into the architecture, which requires pretaining the model from scratch; in contrast, Unlimiformer leverages any already-pretrained model, and thus can be applied to future models as well.

**Other efficient processing methods** Outside of retrieval, many other works have attempted to combine inputs encoded across multiple context windows to process long inputs. This may be

achieved by running the model over sliding windows (and using clustering to permute information between windows) (Wang et al., 2021); by learning a pooling operation over a set of independently-encoded examples (Lee et al., 2019); by attending over clusters of embeddings (Vyas et al., 2020); by learning a retriever to determine a subset of embeddings to attend to (Qin and Durme, 2023); by performing fusion in-decoder (Ivgi et al., 2022); or by using bucketed local attentions with hashing Kitaev et al. (2020). Most of these methods either modify the architecture or introduce additional trainable components.

