# OpenReview forum: "Unlimiformer: Long-Range Transformers with Unlimited Length Input"
_NeurIPS.cc/2023/Conference — NeurIPS 2023 poster_

### Official Review · Reviewer_4zy3 · 2023-07-06

**Soundness:** 2 fair
**Presentation:** 3 good
**Contribution:** 2 fair
**Rating:** 5
**Confidence:** 4

**Summary:**

This paper proposes to use k-nearest neighbor to extract nearest neighbor encoded tokens in pretrained encoder-decoder transformers. This helps in removing the bottleneck of limited input tokens thereby letting the dataset decide the input length. They show empirically that their proposed unlimiformer can extend to beyond 500K and work well with pretrained methods such as BART and Longformer.


**Strengths:**

* The proposed method is a simple add on top of existing pre-trained models
* Equation 2 is a somewhat novel rewrite of QK.
* The authors performed extensive experiments showcasing that a) unlimformer works well on pretrained model, b) in many cases using the full dataset and then selecting the top-k tokens for cross attention results in increase in performance c) the computational increase is less than linear with increasing token size.


**Weaknesses:**

* The idea of retrieving top-K most similar tokens to approximate the softmax has been explored in Reformer. They used LSH to retrieve top-K and that method should work even in the cross attention scenario. In their experiments they have experimented with 64K tokens. In the light of that the method proposed by this paper is not very dissimilar. This makes it hard to recommend accepting the paper. If the authors think that the proposed method is different please comment on it and it should have been compared in the experiments.
* Another thing that is not clear to me is why the forward pass of the *encoder* is not quadratically increasing in time complexity as the number of tokens increase, and why this was not included in Figure 4.


****
These questions were answered during the rebuttal phase

**Questions:**

I will request the authors to please comment on the points raised in the weakness section. I will update my scores accordingly

---

> ### Author Rebuttal · Authors · 2023-08-09
>
> Thank you for taking the time to review our paper! We were happy to read that you appreciated our main points: that Unlimiformer is simple, achieves significant improvements, and allows scaling of the input with less than linear increase in wall-clock time.
>
> We think that your concerns are addressable within this discussion period. Please see our response below. We would love to further address additional questions during the discussion period if anything is unclear.
>
> **"The idea of retrieving top-K most similar tokens to approximate the softmax has been explored in Reformer. They used LSH to retrieve top-K and that method should work even in the cross attention scenario. In their experiments they have experimented with 64K tokens. In the light of that the method proposed by this paper is not very dissimilar. This makes it hard to recommend accepting the paper. If the authors think that the proposed method is different please comment on it and it should have been compared in the experiments."**
>
> We believe Unlimiformer differs substantially from Reformer. Reformer uses bucketed local attentions, where each token can only attend to tokens in the same bucket, and all tokens are attended to at all steps. This means that Reformer requires significant changes to the model architecture and **cannot be applied to existing pretrained models**. By contrast, Unlimiformer:
> * Can be applied to existing pretrained models, since it does not change their architecture
> * Can be applied without any additional training at all (including finetuning), if desired
> * Allows different tokens to be in the same attention window at different steps/heads/layers. In Reformer, tokens can never attend outside of the hash bucket they have landed within.
>
> We compared Unlimiformer to Memorizing Transformers (Wu et al., ICLR’2022), Longformer-Encoder-Decoder (LED; Beltagy et al, 2020) with the summarization-specific pretraining of PRIMERA (Xiao et al, 2022, ACL’2022), and SLED (Ivgy et al., TACL’2023), as we believe these are the most similar approaches. We also note that Longformer demonstrated improvements over Reformer, and we demonstrate empirical and conceptual improvements over Longformer-family models.
>
> However, we do agree that clarifying the differences with existing approaches is quite important, so we will add some of this discussion to the revised draft.
>
> **"Another thing that is not clear to me is why the forward pass of the encoder is not quadratically increasing in time complexity as the number of tokens increase, and why this was not included in Figure 4."**
>
> Because we do not modify the encoder architecture in Unlimiformer, we encode all inputs in overlapping chunks (see section 2.1 in the paper and Figure 1 in the extra 1-page Figures PDF). This means that, as the number of tokens increases, the *number of forward passes of the encoder increases linearly*, with each forward pass having the same time complexity (quadratic over the small, fixed, context window size) as the base model. Specifically, if the model has a context window size of N, each pass of N tokens through the encoder has complexity N^2, where N is fixed. Encoding an input of size 10N with Unlimiformer requires passing 20 chunks of N tokens each through the encoder; encoding an input of size 100N requires passing 200 chunks through the encoder, which is linearly more expensive.
>
> The time to encode the inputs is included in Figure 4– the trend is still sublinear, as the encoding is a minority of the computational cost of the entire model’s prediction, and the cost of decoding scales sublinearly.  Overall, when the input is 100k tokens long, **Unlimiformer is only ~3.5x slower while processing 100x more input** than the base model. This includes the additional time to encode the input.

---

> > ### Comment · Reviewer_4zy3 · 2023-08-15
> > **Response to Author Rebuttal**
> >
> > I thanks the authors for clarifying my questions and for the new results/analysis in the attached pdf. As a result I have moved up my initial score by two points.
> >
> > One request/recommendation for the paper will be to clearly describe their chunking algorithm in the main paper and contrast it with related work

---

> > > ### Author Response · Authors · 2023-08-15
> > >
> > > Thank you for the response and for raising your score!
> > >
> > > We will add additional clarification of the chunking. Thank you for your review, we feel it has helped us improve the paper.

---

### Official Review · Reviewer_VEz6 · 2023-07-06

**Soundness:** 3 good
**Presentation:** 3 good
**Contribution:** 3 good
**Rating:** 6
**Confidence:** 3

**Summary:**

The paper proposes Unlimiformer, a new method for increasing the context length of Transformers without any modification by using retrieval. The idea is simple, and immediately improves performance on benchmark tasks.

**Strengths:**

The idea is simple, and the experiments show that augmenting Transformers with a retrieval system is useful. Inference cost increases sublinearly with the amount of input, which is helpful. Overall a solid idea and execution of the analysis.

**Weaknesses:**

The major weakness is understanding whether the model can actually use all the context that it is being exposed to. It seems clear that using a nearest-neighbor based approach would limit the effectiveness of the context length to some fixed amount of input (once the KNN is "saturated" with nearby elements to the query it does not use further contexts).

Figure 3 seems to suggest that the model does not use data beyond 32K context, or at least has a lot of trouble doing so. The trend is downwards after 32K until the 350K point at the end, which could be an outlier. A natural extension would be to adapt the KNN search in some way as the amount of context increases.

**Questions:**

Do you have more evidence that all the context is used as you increase the context?

**Limitations:**

Yes

---

> ### Author Rebuttal · Authors · 2023-08-09
>
> Thank you for taking the time to review our paper! We were happy to read that you appreciated our main points: that Unlimiformer is simple, immediately improves performance, and does not require any modification to the architecture.
>
> We think that all your questions are addressable within this discussion period. Please see our response below. We would love to further address additional questions during the discussion period if anything is unclear.
>
> **"It seems clear that using a nearest-neighbor-based approach would limit the effectiveness of the context length to some fixed amount of input (once the KNN is "saturated" with nearby elements to the query it does not use further contexts)."**
>
> In our preliminary experiments, we found that taking the 1024-nearest-neighbors covers more than 99% of the full attention mass, so we are not sure that retrieving more keys at the same time is really necessary.
>
> Further, since each attention head in each decoder layer retrieves its own kNN keys, the context length is **not** limited to 1024 or so, since the decoder can attend to `1024 * num_layers * attention_heads` potentially different tokens in each decoding step.
>
> To perform a fair evaluation, we fixed the number of nearest neighbors that we retrieved to be equal to the size of the vanilla model’s context size (e.g., 1024), such that both the base vanilla model and Unlimiformer can attend to the same *number* of keys.
>
>
>
> **"Do you have more evidence that all the context is used as you increase the context?"**
>
> To investigate, we plotted the frequency of retrieval for keys across the full decoding process. In book summarization, we found that the test-time-only Unlimiformer retrieved, on average, 43.5% of the encoded tokens at least once at test time; and the “alternating training” model retrieved 64.5% of the encoded tokens at least once, on average. These were the models with the least and most coverage of the input tokens, respectively. Also note that we retrieve **encoded, contextualized hidden states**, so even vectors that were not directly retrieved by the decoder impacted the final output.
>
> We found no specific skew or pattern in the retrieved keys, and keys from the entire input were used by the model; for all models, the median *location* of a retrieved key was between 49.73% and 49.87% of the way through the input document. See also Figure 2 in the additional 1-page Figures PDF.
>
> We cannot prove that “**all** the context is used”, since there are likely to be parts of each input that are not needed to be used (if they are irrelevant to the output). However, Unlimiformer has the ability to use any part of the context, by retrieving it using a computation that is equivalent to attending to it. The only difference between Unlimiformer and the baselines is the ability to look at the entire output, and Unlimiformer improves base models on long sequences either with or without training.
>
> **"Figure 3 seems to suggest that the model does not use data beyond 32K context, or at least has a lot of trouble doing so"**
>
> We believe that this is mostly a limitation of evaluating such long and information-heavy generations. Although there is a slight drop at 64k and 100k, the performance there is still significantly better than the vanilla BART base, and the general trend is that processing longer context leads to better outputs. Unlimiformer has no inherent preference for the “beginning” or the “end” of a book, and all keys can be equally retrieved.
>
> **"A natural extension would be to adapt the KNN search in some way as the amount of context increases."**
>
> Yes, but we see this as out of the scope of the current work. We believe that the most simple and natural idea is to retrieve the same number of keys as the base model, in order to perform a fair comparison to the base model.
>
> Further, although an interesting potential improvement, setting $k$ to be a larger number than the vanilla model’s context window size would *require* training, because the decoder was not trained to attend to more keys than its context window size; thus, it will take away the advantage of using Unlimiformer training-free.
>
> We thus leave this extension to future work. Thank you for this suggestion.

---

> > ### Comment · Reviewer_VEz6 · 2023-08-11
> >
> > Thank you for the extensive rebuttal and additional experiments. I will be raising my score to a 6.

---

> > > ### Author Response · Authors · 2023-08-11
> > >
> > > Thank you for responding and for increasing your score!
> > >
> > > Please let us know if there are additional questions before the end of the discussion period.

---

### Official Review · Reviewer_cHBL · 2023-07-20

**Soundness:** 3 good
**Presentation:** 2 fair
**Contribution:** 3 good
**Rating:** 7
**Confidence:** 3

**Summary:**

The paper proposes a method to increase context lengths of encoder-decoder transformers to very long input sequences. The idea is to essentially encode all of the tokens of the entire input (on overlapping context-length chunks) and create an index of the input tokens. Just prior to decoding, k-nearest neighbor (kNN) encoded tokens are selected from the index to maximize $QK^T$  dot product of the cross-attention phase, between the encoder and decoder hidden states, during decoding. One of the key contributions in this paper is that they re-order the computation of the matrix products in the cross-attention ($QK^T$) so that creating a single index of the encoder tokens is sufficient, and kNN look-up can be performed efficiently to compute ($QK_{kNN}^T \approx QK_{best}^T$). Additionally, their method has the advantage of being applied to existing pre-trained architectures without the need for additional weights or tuning. It can be applied directly during inference at test time. Applying it during training gives an additional performance boost.

**Strengths:**

* The idea is good and the experimental evidence is strong.
* The evaluation comparisons using “low-cost” tuning methods, i.e. applying this idea just during validation and testing is also clever.
* The proposed method has interesting advantages,

  1. While the idea of using kNN is not new, it has been done nicely and efficiently,
  2.  The method can also be applied to existing pre-trained encoder-decoder models during inference directly.
  3. There's further boost in performance when applied during training.

**Weaknesses:**

No major weaknesses in the overall quality of the contribution. There are minor weaknesses which the authors can address e.g. the clarity of the writing can be improved, results in some tables are incomplete etc. I have framed those as questions below for the authors to address.

**Questions:**

1. The Fig. 2 shows encoder encoding disjoint chunks, whereas text lines L73-74 say it is overlapping chunks. Can you clarify which of these is correct, and change the figure to represent that more accurately? The figure, overall, can be improved.

2. For the k nearest neighbor indexing and search, you mention the use of the FAISS library. Do you use approximate nearest neighbors, or the exact nearest neighbors? Considering most contexts are fairly small, if you’re not using exact nearest neighbors, can you provide a small comparison to using exact nearest neighbors, and share whether or not it affects results (and/or time).

3. In the Comparison to Wu et. al. section, it would be good to highlight that Memorizing Transformer paper applies their approach to decoder-only models. Your approach is applied to the encoder-decoder model.

4. Train chunked +test Unlimiformer (L137-141) requires more details. e.g., I don’t understand what treating each chunk as its own training example entails. Is this what you do: For an example that’s longer than the context length, do you chunk it into say C non-overlapping chunks, treat each (chunk, output) pair as a training example (to create C examples), but allow retrieval over the tokens of all the C chunks during decoding? (After reading retrieval training, *I guess there is actually no retrieval happening over training data* in the Train chunked +test Unlimiformer)

  * 4.1 For Train chunked +test Unlimiformer, how does the duplication of some training examples because of the C (chunk, output) pairs figure into evaluation? Do you try to combine the C outputs to a single one, or take the average, or ignore the duplication and just treat them as more example pairs than what other methods would have? (Just clarify this detail in the paper)

5. Is Sec. 3.2 also doing fine-tuning? I was quite confused by the statement “We also consider training Unlimiformer directly”. Perhaps a different way to separate out section 3.1 and 3.2 is to say that in 3.1. you are applying kNN look up during model evaluation (while evaluating the models after each period of training), Vs 3.2 where you are applying kNN strategy on the training data examples (where the choice of the kNN values directly affects the model gradients during tuning).
6. During retrieval training, for the training examples that are longer than the context length, do you just have 1 copy of the (input, output) pair example in training – unlike the train chunked case where you could have multiple copies? Is this true even for training examples that are longer than 16k tokens?
7. In Table 4 when you say “Unlimiformer (this work)” do you mean the “Retrieval training” regime?
8. In Table 5 for BookSum, why isn’t there a line reporting results for BART-base Unlimiformer (Retrieval Training)?
9. Table 4 doesn’t directly include PRIMERA base training (standard finetuning) results, but worth repeating that result line from Table 3, so that this sentence (L206-207) is easier to follow: “in Table 4 highlight two important points: first, Unlimiformer+BARTbase performs better than the base PRIMERA across all metrics and datasets”
10. In the Comparison to SLED Ivgi et. al. The statements in L274-276
*“…This in practice limits SLED to only about 16k token-long inputs on a single GPU; in contrast, instead of attending to all input tokens, Unlimiformer attends only to the top-k input tokens for every attention head, and thus can process unlimited inputs in practice”*
Contradicts your statements in Sec. 3.2 Retrieval Training L149-150
*“When inputs are longer than 16k tokens, we truncated the input to 16k tokens at training time due to GPU memory requirements.”*
So, it seems like atleast in practice, in the experiments reported in this paper the input length was restricted to max 16k tokens due to practical limitations. Is that correct to say?
11. [Comment] In Table 2 (Sec 4.1. Datasets), for the distribution plots it might be worth adding a vertical line to indicate where the average length (avg. number of input tokens) falls in the distribution curve.

**Limitations:**

Perhaps worth noting that this is not applied to decoder-only models.

Also worth noting the 16k limitation to fit into GPU during training.

---

**Post rebuttal notes**

The authors rebuttal and subsequent discussion addressed all my questions and concerns.

---

> ### Author Rebuttal · Authors · 2023-08-09
>
> Thank you for taking the time to review our paper! We were happy to read that you appreciated our main points: that Unlimiformer can be applied to existing pre-trained architectures without the need for additional weights or tuning, our re-order of the computation of cross-attention can use a single index of the encoder tokens, and that Unlimiformer can be applied directly at test-time.
>
> We are happy to hear that you think that there are no major weaknesses. We think that all your questions are addressable within this discussion period. Please see our response below. We will include these clarifications in the paper, and we would love to further address additional questions during the discussion period if anything is unclear.
>
>
> **Q1. Fig. 2 shows encoder encoding disjoint chunks, whereas text lines L73-74 say it is overlapping chunks. Can you clarify and change the figure to represent that more accurately?**
>
> We encoded overlapping chunks to have sufficient context, and then we took only the “middle” of each encoded chunk, such that eventually we have a single encoded vector for each input token. Thank you for this suggestion, we improved the figure, and the new version is now included as Figure 1 in the 1-page Figures PDF.
>
> **Q2. For the k nearest neighbor indexing and search, do you use approximate nearest neighbors, or the exact nearest neighbors? Can you provide a small comparison to using exact nearest neighbors, and share whether or not it affects results (and/or time)?**
>
> In GovReport and SummScreen, where the validation inputs are 20k-70k tokens long at most, we tried using exact nearest neighbors (a “flat” faiss index). However, using approximate nearest neighbors did not change the results **at all**.
> Querying the index using approximate-NNs is much faster; however, it takes a few seconds to build the initial index for each example, so the time to process an entire test example was very similar overall.
>
> **Q3. In the Comparison to Wu et. al. section, it would be good to highlight that Memorizing Transformer paper applies their approach to decoder-only models. Your approach is applied to the encoder-decoder model.**
>
> We agree, and we will clarify this in the paper.
>
> **Q4. Train chunked +test Unlimiformer (L137-141) requires more details [...]; Q4.1 how does this figure into evaluation?**
> The first part of your description is correct: “chunk it into say C non-overlapping chunks, treat each (chunk, output) pair as a training example (to create C examples).” As you noted, we do not do retrieval during chunked training– the examples are treated as completely distinct. The idea in this chunking is to try to achieve the best results **without** increasing hardware requirements beyond the standard finetuning.
> At test time, we take `(full input, output)` as a single training example, without using chunking at all, and apply Unlimiformer to process the entire input. Thus, chunking is a data augmentation choice at training-time that is independent of the model used. We will clarify.
>
>
> **Q5. Perhaps a different way to separate out section 3.1 and 3.2 is to say that in 3.1. you are applying kNN look up during model evaluation, vs 3.2 where you are applying kNN strategy on the training data examples**
>
> Your rephrasing here is correct and much clearer-- thank you for the suggestion!
>
> **Q6. During retrieval training, for the training examples that are longer than the context length, do you just have 1 copy of the (input, output) pair example in training?**
>
> Yes, we have only a single copy of each example in retrieval training. In the case where examples are longer than 16k tokens, we could employ chunking with chunk size 16k, but we found that the method worked well in practice without this additional step.
>
>
>
> **Q7. In Table 4 when you say “Unlimiformer (this work)” do you mean the “Retrieval training” regime?**
>
> This is the “alternating training” (alternating batches of “retrieval training” and “random-encoded training”) which worked best here; we will clarify this in the paper.
>
>
> **Q8. In Table 5 for BookSum, why isn’t there a line reporting results for BART-base Unlimiformer (Retrieval Training)?**
>
> We apologize for the omission– we will include these results in the next version, and in the 1-page Figures PDF (see Table 1).
>
> **Q9. Table 4 doesn’t directly include PRIMERA base training (standard finetuning) results, but worth repeating that result line from Table 3**
>
> “PRIMERA standard finetuning” is presented as “LED-large - PRIMERA” in Table 4. We agree that this is unclear, and we will clarify this.
>
> **Q10. it seems like at least in practice, in the experiments reported in this paper the input length was restricted to max 16k tokens due to practical limitations. Is that correct to say?**
>
> SLED’s limitation is during both training and inference; Unlimiformer has a practical limitation on length at training-time, but no practical length restriction at test-time. In all Unlimiformer results in the paper, we are using the full test-set inputs without any truncation at test time.
>
> **Q11. [Comment] for the distribution plots: add a vertical line to indicate the average length**
> This is a great idea! We included this suggestion in Table 2 in the 1-page Figures PDF.
>
> **Perhaps worth noting that this is not applied to decoder-only models.**
>
> In the scope of this paper, we focused on encoder-decoder models. We will clarify this.
>
> Although we did not perform an extensive evaluation with LLama-2 (and thus it is out of the scope of the paper), our codebase (which we will publicly release) recently supports decoder-only models as well, such as LLama-2: in decoder-only models, we need to keep an index for each layer, but we can still leverage our Attention Reformulation (Section 2.3) to share the same index among all attention heads in the same layer.
>
> **Also worth noting the 16k limitation to fit into GPU during training.**
>
> Thank you, we will include this in the Limitations section.

---

> > ### Comment · Reviewer_cHBL · 2023-08-11
> > **Any evals on tasks other than summarization?**
> >
> > Thanks for your responses. it's good to also know about the extension to decoder-only models.
> >
> > Have you evaluated on tasks other than summarization? Even if the inputs were not really long?
> >
> > I agree with the concern raised by Reviewer ym6S, regarding eval on summarization tasks alone. Despite your response there, it does seem like a limitation. It would be interesting to know if you have done any evals on one or 2 other tasks, perhaps the others from the SCROLLS benchmark? This might be particularly relevant for the models that have been trained with the modified retrieval layer (retrieval/enc-dec/alternating) training. Even if it is on tasks where the input length is not large, it would be of interest to validate whether the model performance regresses on those tasks (and if it does to what extent, if it doesn't that's also useful to know).

---

> > > ### Author Response · Authors · 2023-08-14
> > >
> > > Thank you for your response!
> > >
> > > We have queued experiments on more tasks from the SCROLLS benchmark, and we will report these results in the next few days.

---

> > > > ### Author Response · Authors · 2023-08-18
> > > > **Additional SCROLLS results**
> > > >
> > > > >It would be interesting to know if you have done any evals on one or 2 other tasks, perhaps the others from the SCROLLS benchmark? ... it would be of interest to validate whether the model performance regresses on those tasks
> > > >
> > > > Thanks to your suggestion, we ran experiments on 4 additional QA and NLI tasks from SCROLLS. Please see the global response!

---

> > > > > ### Comment · Reviewer_cHBL · 2023-08-20
> > > > > **Nice work!**
> > > > >
> > > > > Thanks for providing the additional information,  clarifications, and the additional experimental results.
> > > > >
> > > > > Nice work!

---

### Official Review · Reviewer_ym6S · 2023-07-27

**Soundness:** 3 good
**Presentation:** 3 good
**Contribution:** 3 good
**Rating:** 7
**Confidence:** 4

**Summary:**

This paper proposes to use kNN based search to replace the notoriously memory consuming quadratic attention in modern Transformers to allow extremely long sequence input. Proposed method is simple, and can be applied to any pre-trained Transformer. The proposed model, Unlimiformer, is evaluated on long text summarization tasks, and achieves significant improvement over baselines due to the increased context length. Note that the proposed method are shown to be effective without any finetuning on some tasks.

**Strengths:**

S1. Sequence length is a major pain point of modern Transformer. This work targets a significant issue and achieves promising results.

S2. Proposed method can be applied to multiple pre-trained models (BART, PRIMERA)

**Weaknesses:**

W1. Only evaluated on text summarization.

W2. Memory / Speed trade-offs are not quantitatively studied.

Evaluation on one single task is the major limitation of this work. The usage of Transformer is broad. For example, long document QA tasks should also be studied. Long Range Arena is also a useful benchmark for efficient Transformers. Beyond NLP, I am also curious if this can be applied to computer vision transformers such as ViT, to enable higher-resolution input images.

In addition, memory / speed trade-offs could benefit the readers for clearer guidelines when applying the proposed method. I strongly suggest adding this study. I am also curious how this method scale to LLMs larger than BART.

Currently, I think the strengths outweigh the weaknesses so I'm leaning toward acceptance.

--------------
update after rebuttal
===============
Both W1 and W2 are properly addressed by the authors. Additional results on SCROLLS shows that Unlimiformer can generalize to other NLP tasks beyond summarization and an additional pre-trained model T5 is also studied. The additional memory / speed study also demonstrate the advantage to the proposed method. Since this method can be applied to many tasks and many pre-trained transformer LMs, it has potential to achieve high impact in the community. I decided to increase my rating from 5 to 7. My confidence score is also raised from 3 to 4. I think this paper is a clear accept. The reason I'm still a little conservative is that I don't have first hand experience in working on these specific benchmarks.

**Questions:**

Suggestions:
- I would tone down the claim of 'unlimited length', because CPU memory is, although typically larger than GPU memory, still practically limited. Instead, I suggest the authors to stress test the limit for a specific hardware (CPU&GPU) with synthetic long sequences so users have a practical reference.
- Consider citing more related works such as Cluster-Former [1], Set Transformer [2], and Fast Transformer [3].

--------------
update after rebuttal
===============
I don't have further questions and suggestions for the submission. Thanks for the response!

[1] Wang et al., "Cluster-Former: Clustering-based Sparse Transformer for Question Answering", Findings of ACL 2021

[2] Lee et al., "A framework for attention-based permutation-invariant neural networks", ICML 2019

[3] Vyas et al., "Fast transformers with clustered attention", NeurIPS 2020

**Limitations:**

Limitations are properly addressed.

---

> ### Author Rebuttal · Authors · 2023-08-09
>
> Thank you for taking the time to review our paper! We were happy to read that you appreciated our main points: that Unlimiformer is simple, achieves significant improvements, is effective even without any finetuning, and that you think that the strengths outweigh the weaknesses.
>
>
> We think that all your questions are addressable within this discussion period. Please see our response below. We would love to further address additional questions during the discussion period if anything is unclear.
>
> **"The approach is evaluated only on text summarization [...] I am also curious if this can be applied to computer vision transformers such as ViT..."**
>
> We focused on summarization because producing summaries with high ratios of compression requires using information from full, sometimes extreme-length inputs.
> Thus, our evaluation includes long document summarization and book summarization, using 3 datasets (GovReport, SummScreen, BookSum) and 2 base models (BART, LED).
>
> We are happy to hear that our approach sparks ideas for additional modalities, and we agree that it would be interesting to apply this approach to vision transformers, but we think that this is a bit out of the scope of the current paper, as we focus on natural language transformers.
>
> We also have results for T5 as the base model, which we will include in the next version.
>
> **"Speed trade-offs are not quantitatively studied"**
>
> Processing extremely long inputs with Unlimiformer is not as “real-time” fast as vanilla Transformers yet, but we study speed trade-offs in Figure 4 and Section 6.
> For example, when the input is 100k tokens long, **Unlimiformer is only ~3.5x slower while processing 100x more input** than the base model. This includes the additional time to encode the input.
>
> **CPU memory is, although typically larger than GPU memory, still practically limited. Is Unlimiformer really unlimited?**
>
> Of course, nothing in nature is unlimited. However, we argue that Unlimiformer is effectively unlimited since it can process any practical input, since Unlimiformer requires keeping only a single vector per input token. For example, using BART or T5-Large, where hidden states are of size 1024, encoding **1M tokens** using fp16 takes **only 2GB of memory**, which can fit multiple times on GPU memory, let alone on CPU memory. Additionally, if CPU memory is truly insufficient, the kNN implementation we used (faiss) supports saving indices on disk and loading only portions of them into memory at a time for search (see their documentation for details).
>
> **"I am also curious how this method scale to LLMs larger than BART."**
>
> Most of our experiments were conducted using BART-base or PRIMERA (Longformer-Encoder-Decoder-large) as the base models.
> After the submission deadline, we managed to run experiments with T5 (base) on GovReport as well:
>
> | Model | Rouge 1 / 2 / L / BERTScore |
> |-----------------|-----------------------------|
> | T5 - standard finetuning    |       41.9 / 17.0 / 26.7 / 61.2                      |
> | T5 - Unlimiformer (alternate training)      |  **50.1** / **22.8** / **26.4** / **64.3** |
>
> We will add this result to a revised version of the paper.
>
> Although it’s out of the scope of our paper, our codebase supports LLama-2 and we managed to encode and summarize entire books (hundreds of thousands of tokens) using Unlimiformer-LLama-2 (13B) within few minutes, using only 2 A6000 GPUs.
> We thus believe that in terms of scaling, our approach scales very well to larger models as well.
>
>
> **"Consider citing more related works such as Cluster-Former [1], Set Transformer [2], and Fast Transformer [3]."**
>
> Thank you, we will cite and discuss these works in our revised version.

---

> > ### Comment · Reviewer_ym6S · 2023-08-14
> > **Reviewer's response**
> >
> > Thanks for responding to my questions and concerns. I hope my comments / suggestions are helpful for your future revisions.
> >
> > For the speed-memory tradeoff (W2), perhaps I wasn't clear in my original review. I was hoping to see some memory - speed plot when replacing 0, 1, 2, ... to all layers in the baseline Transformer with an UnlimiFormer layer under a fixed sequence length. Fig. 4 is good, but I'm just suggesting more experiments for a more complete study.
> >
> > I understand that the authors focus on summarization and this is well motivated. However, this remains the main reason (W1) that I won't raise my final rating. A lot of efficient transformers have been proposed, and the most impactful ones are those experimented on multiple NLP tasks. For a task-specific method like this submission, perhaps the authors should refrain from using such a generalist paper title. Perhaps "Long-Range Transformer for Unlimited-Length Summarization" better fits the current scope.

---

> > > ### Author Response · Authors · 2023-08-14
> > >
> > > Thank you for your response!
> > >
> > > **For W2:** Apologies for the misunderstanding! We have now run the analysis you proposed on BART-base+Unlimiformer (retrieval).
> > >
> > > | Number of layers with Unlimiformer | Max GPU memory allocated | Total time taken | Entity Mention (EntMent) |
> > > | ---------------------------------- | ------------------------ | ---------------- | ------- |
> > > | 0 (normal inference)               | 24.8%    (11.9 GB)       |   6m27s          |    9.8  |
> > > | 1                                  | 85.5%    (41.0 GB)       |  16m37s          |   15.3  |
> > > | 2                                  | 89.6%    (43.0 GB)       |  19m32s          |   15.2  |
> > > | 3                                  | 85.9%    (41.2 GB)       |  19m35s          |   17.7  |
> > > | 4                                  | 86.1%    (41.3 GB)       |  21m25s          |   17.3  |
> > > | 5                                  | 84.1%    (40.4 GB)       |  22m53s          |   18.8  |
> > > | 6 (all layers)                            | 85.2%    (40.9 GB)       |  26m14s          |   20.3  |
> > >
> > >
> > > Here, we are measuring maximum GPU memory allocated, total time, and Entity Mention over the full BookSum test set using a single A6000 GPU. The longest example in the test set is approximately 505k tokens.
> > >
> > > The total time taken does increase when we apply Unlimiformer to more layers, but the memory allocated does not, because we use our attention reformulation to reuse the same datastore across layers. We see improved Entity Mention (EntMent) when we apply Unlimiformer on more layers, and applying Unlimiformer at all layers requires only ~3.4x more memory and ~4x more time than running the base model while processing inputs that are 100-500x longer.
> > >
> > > (The slight differences in max memory allocated are because the profiler we used is taking snapshots at slightly different times, not because there is significant variation in memory usage between 1-6 Unlimiformer layers. We will average over repeated runs for stability when we report these results in the paper).
> > >
> > > **For W1:** We have queued experiments on more tasks from the SCROLLS benchmark, and we will report these results in the next few days.

---

> > > > ### Comment · Reviewer_ym6S · 2023-08-16
> > > > **Reviewer's response to additional time/memory table**
> > > >
> > > > Thanks for the response. I also look forward to additional results on SCROLLS.
> > > >
> > > > The speed/memory table is interesting. Could you further clarify where did the additional memory come from (from baseline to 1 layer)? Is the baseline's input length truncated? I thought the proposed kNN approach would have some time / memory saving but they both decayed from the table.

---

> > > > > ### Author Response · Authors · 2023-08-18
> > > > > **Corrected memory consumption and additional SCROLLS results**
> > > > >
> > > > > We realized that we made a mistake in our previous memory profiling. We re-ran our code, and additionally changed to `fp16`. Here are the updated memory consumption results:
> > > > >
> > > > > | Number of layers using Unlimiformer | Memory consumption (GB) |
> > > > > | ---------------------------------                    | -------------------- |
> > > > > |	0 (normal inference) |	1.61	|
> > > > > |	1	|	7.33	|
> > > > > |	2	|	7.32	|
> > > > > |	3	|	7.36	|
> > > > > |	4	|	7.32	|
> > > > > |	5	|	7.33	|
> > > > > |	6 (all)	|	7.35	|
> > > > >
> > > > >  >The speed/memory table is interesting. Could you further clarify where did the additional memory come from (from baseline to 1 layer)? Is the baseline's input length truncated?
> > > > >
> > > > >  As you mentioned, there is a slight increase between the base model and using Unlimiformer in 1 layer.
> > > > >  There are two reasons for this:
> > > > > * there is a slight overhead for constructing the index and querying it
> > > > > * more crucially, the base model here ("normal inference") is truncating the input to the first 1024 tokens. When Unlimiformer is used, we process the full inputs, some of which are >500,000 tokens, and so much of the additional memory cost comes from storing the additional ~499k hidden states.
> > > > >
> > > > >  However, the highlight of this table to us, is to see how GPU memory consumption remains **constant** even when we use Unlimiformer in more layers. This highlights the scalability of Unlimiformer compared to other approaches such as Memorizing Transformers, which need to allocate more memory with every layer and every attention head. Because this previous work required 2 datastores per head and BART has 12 attention heads, it would require **~60GB** to augment a single layer with retrieval, and **hundreds** of GB to augment all layers with retrieval. Because of this, previous methods were unable to perform retrieval at more than one layer.
> > > > >
> > > > >
> > > > > >I also look forward to additional results on SCROLLS.
> > > > >
> > > > > Thanks to your suggestion, we ran experiments on 4 additional QA and NLI tasks from SCROLLS. Please see the global response!

---

> > > > > > ### Comment · Reviewer_ym6S · 2023-08-18
> > > > > >
> > > > > > Thanks for the detailed response and additional results! Since all my concerns are properly addressed by the authors, I'll raise my final score from 5 to 7.
> > > > > >
> > > > > > I strongly encourage the authors to further refine the additional experimental results conducted during the rebuttal period, and then discuss them in details in the next paper revision.

---

> > > > > > > ### Author Response · Authors · 2023-08-18
> > > > > > >
> > > > > > > Thank you for engaging in discussion and for raising your score!
> > > > > > >
> > > > > > > We will definitely incorporate the experiments on the additional tasks and more discussion of memory requirements in the next version.

---

### Official Review · Reviewer_EQX8 · 2023-08-03

**Soundness:** 3 good
**Presentation:** 3 good
**Contribution:** 3 good
**Rating:** 7
**Confidence:** 4

**Summary:**

Unlimiformer: Transformers with Unlimited Input Length:- proposes a novel method to overcome the context window limitation in encoder-decoder Transformer models. The key innovation introduced in this paper is a retrieval-based method that integrates a k-Nearest Neighbors (kNN) search into each decoder layer of the model. This approach allows each attention head to choose different context windows from the full-length input at each decoding step.

The authors propose an encoding process for long input sequences that uses the model's encoder on overlapping chunks of the input. To ensure sufficient context, only the middle half of the encoded vectors from each chunk are kept.

The authors diverge from the standard Transformer cross-attention mechanism by retrieving the top-k hidden states from the entire input sequence. The proposed method involves a mathematical reformulation of the attention mechanism to incorporate the kNN search into the cross-attention process more efficiently.

**Strengths:**

This approach can be integrated into any existing pretrained encoder-decoder transformer model to enable summary over unbounded inputs (subject to computational constraints).

This approach does not require retraining of the model, although further finetuning appears to improve performance.

The paper is well written and clear.

**Weaknesses:**

The computational cost of encoding the entire input could be very high at inference time.

Many of the chosen benchmark approaches appear to be pretty arbitrary and not very convincing.

It is unclear to what extent the overlapping approach to encoding inputs aids the approach. Text is order dependent and filler words serve a purpose, which in many cases affects the semantic meaning of a particular word or sentence. A kNN approach over the most relevant words across the entire inputs could just find correlated tokens that lack any local context in how they were used within a particular phrase or sentence..

**Questions:**

How is k chosen? is it always set to the base models context window length? What happens as k is varied?

It would be good to have a better idea of the computational performance of this kNN-based approach and how it scale. Especially in terms of compute, latency and memory. It seems that it would be prohibitively expensive to query the datastore at each timestep?

What is the motivation behind the training methods shown in section 3.2.? How does it weakly simulate nearest neighbours?

Whenever there is mention of surprising results with regard to the model being able to perform well with limited context, or the full input being unnecessary to produce a summary. How confident are you that the base model has not been trained on WikiSum or any of the datasets (and its derivatives) that it is being evaluated on?

In figure 3, what causes the dip at 64k and 100k datastore?

By CPU datastore, do you just mean system RAM? if so, please mention that.

Could you provide examplers of what actually gets retrieved by kNN? As in, the parts of the input that are being used by the model for particular queries?

**Limitations:**

No broader impact statement is provided, but its conclusion would strengthen the paper

---

> ### Author Rebuttal · Authors · 2023-08-09
>
> Thank you for taking the time to review our paper! We were happy to read that you appreciated our main points: Unlimiformer can be integrated into existing pretrained encoder-decoders, makes it possible to summarize unbounded inputs, and does not require retraining.
>
> We think that all your questions are addressable-- please see our response below. We would love to discuss further if anything is unclear.
>
> **The computational cost of encoding the entire input could be very high.**
>
> In order to reason over the entire input, we need to process this input *somehow*. A cost of the base model's encoding cost scaled linearly with the input’s length is comparable to other methods in the literature (e.g. SLED, Memorizing Transformers).
>
> **"Many of the chosen benchmark approaches appear to be pretty arbitrary"**
>
> For datasets - we evaluated Unlimiformer on GovReport and SummScreen, which are the two main summarization datasets from the SCROLLS benchmark (Shaham et al., EMNLP’2022). To evaluate over *even longer* inputs, we included an evaluation on BookSum (Kryscinski et al., EMNLP’2022) as well.
>
> For baselines - we are not aware of any other long-range transformers that can utilize pretrained models, and can process long inputs without re-training. We thus compared Unlimiformer to the most relevant and related models: Memorizing Transformers (Wu et al., ICLR’2022), Longformer-Encoder-Decoder (LED; Beltagy et al, 2020) with the summarization-specific pretraining of PRIMERA (Xiao et al, 2022, ACL’2022), and SLED (Ivgy et al., TACL’2023).
>
> **Does the overlapping approach to encoding inputs help?**
>
> On average, encoding inputs in overlapping chunks led to a slight increase in performance (about 0.5 / 0.5 / 0.4 R1/R2/RL).
>
> **If you retrieve individual words, they might be retrieved out of their local context**
>
> Note that we retrieve **encoded hidden states** (vectors), from the output of the top layer of the encoder, rather than retrieving raw tokens.
> These retrieved vectors are already-contextualized by the encoder, and are thus “aware” of their local context.
>
> **How is k chosen?**
>
> We fixed the number of nearest neighbors that we retrieved to be equal to the size of the vanilla model’s context size (e.g., 1024), such that  Unlimiformer fully replaces the cross-attention to encoder hidden states. Setting k larger than this would require training, because the decoder was not trained to attend to more keys than its context window size. We leave this for future work.
>
> **How does it scale in terms of compute, latency and memory?**
>
> When the input is 100k tokens long, **Unlimiformer is only ~3.5x slower while processing 100x more input** than the base model (see Figure 4 in the main paper). There is some fixed additional cost to construct an index (a few seconds), but decoding is quite fast because retrieval from the index is sub-linear.
>
> The additional memory needed scales linearly, but with a small cost per token, since Unlimiformer requires keeping only a single vector per input token. Using hidden states of size 1024, encoding **1M tokens** using fp16 takes **only 2GB of memory**. The index containing these hidden states can be stored in the GPU memory or in RAM, depending on the compute availability.
>
> **What is the motivation behind the training methods in section 3.2.?**
>
> “Retrieval Training” is a training approach that simulates the test time computation exactly: every cross-attention head performs kNN-search and attends to the retrieved keys. However, we believe that this training approach makes every cross-attention head attend only to “relevant keys” at training time, and thus cross-attention never learns to *down-weight* irrelevant keys. Thus, “Random-Encoded training” selects a random subset of $k$ encoded tokens for each cross-attention head at training time (to expose heads to "irrelevant" keys), while applying kNN at test time. “Alternating training” alternates batches of these methods.
>
> We believe that these motivations were not explained well enough in the paper, and we will include them in the revised version. Thank you!
>
> **How confident are you that the base model has not been trained on the datasets?**
>
> This is a great question about “leakage” of test data from the pretraining data. In WikiSum, we agree that most models are likely pretrained on the entire English Wikipedia, and this is another explanation for the surprisingly strong performance of the baseline here. We will add this to the paper.
>
> The other datasets are not explicitly included in BART’s pretraining data, especially not as pairs of documents/books and their summaries, though it is possible that some were indirectly included through OpenWebText. Although we believe that there is no serious leakage, any leakage would only make the baseline model’s performance be closer to Unlimiformer.
>
> **In figure 3, what causes the dip at 64k and 100k datastore?**
>
> We believe that this is mostly a limitation of evaluating such long and information-heavy generations. Although there is a slight drop at 64k and 100k, performance there is still significantly better than the vanilla BART base. The general trend is that processing longer context leads to better outputs.
>
> **By CPU datastore, do you mean system RAM?**
>
> Yes-- at inference time, we stored the hidden state vectors in a datastore constructed with the library faiss, either in GPU memory (VRAM) or RAM.
>
> **What actually gets retrieved by kNN?**
>
> Note that we retrieve **encoded hidden states** (vectors) rather than tokens, so even vectors that were not directly retrieved by the decoder impacted the final output. When we plotted the frequency of retrieval for keys across the full decoding process in BookSum, we found no strong skew or pattern in the retrieved keys, and keys from the entire input were used by the model (Figure 2 in the rebuttal PDF).
>
> **A broader impact statement would strengthen the paper**
>
> We will include one in the next version-- thank you for the suggestion!

---

### Author Rebuttal · Authors · 2023-08-09

We thank the reviewers for their time and feedback! We are encouraged that all reviewers have noted the benefits of Unlimiformer, including that it can be applied to pretrained models with no additional training, has sublinear inference time w.r.t. the length of the input, and leads to significant performance improvements.

We have provided more details and additional analysis of our results in the responses to each individual reviewer. In the rebuttal doc, we have provided:
* Figure 1: an updated version of the paper's Figure 2, using feedback from reviewer cHBL.
* Figure 2: a new figure demonstrating the locations retrieved from during decoding, to address questions from reviewers EQX8 and VEz6.
* Table 1: an updated version of the paper's Table 5 with results for additional Unlimiformer settings on BookSum, as reviewer cHBL requested.
* Table 2: an updated version of the paper's Table 2 with updated visualizations for dataset input lengths, using feedback from reviewer cHBL.

We look forward to answering any additional questions in the discussion period.

---

### Author Response · Authors · 2023-08-18
**Additional results: QA and NLI**

We would like to thank the reviewers for all their excellent suggestions and for engaging in a discussion! We really feel that they have helped us improve the paper.

Following Reviewer `cHBL` and Reviewer `ym6S`'s suggestions, we trained Unlimiformer and the base model on additional tasks - long-input QA and NLI tasks from SCROLLS.

| Base model | Training        | QASPER (F1) | Contract NLI (Exact Match) | QMSum (ROUGE 1 / 2 / L / BERTScore F1) | Narrative QA |
|------------|-----------------------------------------|-------------|----------------------------|----------------|--------------|
| BART       | Standard finetuning (first 1024 tokens) | 22.0        | 77.5                |   **30.8 / 8.7 / 20.8 / 56.8**  | 15.5         |
| BART       | Unlimiformer (Retrieval Training)       | **27.3** ($\uparrow$5.3)       | **78.8** ($\uparrow$1.3)  |   29.6 / 7.7 / 19.4 / 55.7     | **18.7** ($\uparrow$3.2) |
| BART       | Unlimiformer (Alternating Training)     | 27.2 ($\uparrow$5.2)       | 77.7  ($\uparrow$0.2)  |   29.3 / 7.8 / 19.7 / 55.9      | 18.5  ($\uparrow$3.0)       |

To keep Unlimiformer generic, we did not perform any architectural modifications, and we treated these as sequence-to-sequence tasks: we crafted the inputs by concatenating `question + document` (or `hypothesis + document` in Contract NLI), to represent all tasks as text-to-text. Further, we did not perform *any hyperparameter tuning*: we took hyperparameter values such as learning rate from the SCROLLS benchmark to train the base model, and then used the same hyperparameters to train Unlimiformer.

As shown, both Unlimiformer training approaches significantly improve over the base model in 3 of the 4 tasks, without any task-specific modeling or hyperparameter tuning. We believe that tuning or task-specific tricks could further increase Unlimiformer’s gain, but these results show that even the most straightforward application of Unlimiformer provides benefits across different tasks.

---

### Decision · Program_Chairs · 2023-09-21

**Decision:**

Accept (poster)

**Comment:**

All reviewers agree this paper nicely extends the long context inference for Transformers line of work and showcases strong results. Initial concerns were around limited evaluation and efficiency of the proposed approach. Authors presented additional experimental results during discussion that has addressed most of the reviewers concerns. I am happy to suggest acceptance and encourage authors to update the final draft as per the reviewers suggestions.